# GENERATIVE MONOCULTURE IN LARGE LANGUAGE MODELS

**Fan Wu**[1], **Emily Black**[2†], **Varun Chandrasekaran**[1†]
[1] University of Illinois Urbana-Champaign   [2] New York University
[†] Equal advising
{fanw6,varunc}@illinois.edu, emilyblack@nyu.edu

## ABSTRACT

We introduce *generative monoculture*, a behavior observed in large language models (LLMs) characterized by a significant narrowing of model output diversity relative to available training data for a given task: for example, generating only positive book reviews for books with a mixed reception. While in some cases, generative monoculture enhances performance (e.g., LLMs more often produce efficient code), the dangers are exacerbated in others (e.g., LLMs refuse to share diverse opinions). As LLMs are increasingly used in high-impact settings such as education and web search, careful maintenance of LLM output diversity is essential to ensure a variety of facts and perspectives are preserved over time. We experimentally demonstrate the prevalence of generative monoculture through analysis of book review and code generation tasks, and find that simple countermeasures such as altering sampling or prompting strategies are insufficient to mitigate the behavior. Moreover, our results suggest that the root causes of generative monoculture are likely embedded within the LLM's alignment processes, suggesting a need for developing fine-tuning paradigms that preserve or promote diversity.

## 1 INTRODUCTION

Large language models (LLMs) show promise due to their emergent abilities (Wei et al., 2022a) and state-of-the-art performance on several NLP tasks (Bommasani et al., 2023). However, concerns have been raised about the increasing reliance on LLM-based systems with insufficient testing, and how they impact society (Anwar et al., 2024; Wang et al., 2023). Recent evidence has shown that LLMs have dangerous tendencies: they convincingly return incorrect information (Dahl et al., 2024; Li et al., 2023a; Zhang et al., 2023), produce toxic language (Abid et al., 2021; Wen et al., 2023), and can effectively propagate misinformation (Barman et al., 2024; Sun et al., 2024).

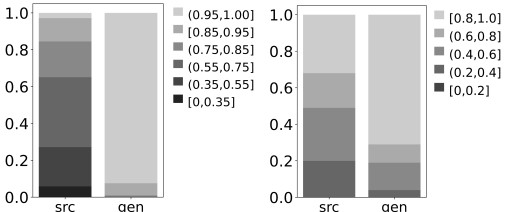

Figure 1: **(Left)** Comparison of the range of average-per-book sentiment scores for book reviews generated by an LLM (gen) and by human reviewers from the Goodreads dataset (src). *Note the generated reviews have a much smaller range, as they are overwhelmingly positive.* Model: Llama-2-chat. **(Right)** The spectrum of the mean pairwise Jaccard similarity among the algorithms of coding solutions. Note the generated code covers a narrower range of algorithms. Model: GPT-4

In this paper, we focus on a different concern: that for a given prompt and task, *LLMs do not faithfully represent the diversity of potential responses available in their training data*. We call this behavior *generative monoculture*: given some task (e.g., generating book reviews) and a data attribute (e.g. sentiment), generative monoculture refers to the narrowing of the probability distribution of the considered attribute from source data (i.e., available human-written book reviews as part of the training data) to the generated data (i.e., LLM-generated book reviews).

As a preview, for book reviews (Fig. 1(**Left**)), we compare the diversity in sentiment of Goodreads reviews (Wan et al., 2019) (i.e., src)—very likely a portion of LLM training data (Achiam et al., 2023)—with LLM-generated reviews (i.e., gen). The range of mean sentiment scores per book

across `gen` book reviews is much narrower than that in the `src`: in the experiment pictured, the average sentiment score for `gen` reviews are mostly over 0.85, whereas the the average sentiment score for `src` reviews over the same books have a wider range from zero to one. For the code generation task (Fig. 1(**Right**)), the range of algorithms employed in (correct) solutions to a given coding problem (i.e., `gen`) was much less varied than a sample of human answers (`src`) available on the web (Li et al., 2022): we show this through the range of Jaccard similarity of the algorithms employed in sets of human-written (`src`) and LLM-generated (`gen`) responses to a coding prompt.

Through the rapid adoption of LLMs such as ChatGPT, CoPilot, and Devin across education, code generation, and day-to-day information gathering, generative monoculture can harm society through loss of information, creativity, and intellectual diversity. For example, students asking LLMs questions about class material to get help researching for an essay may have their opinions formed without exposure to a sufficiently wide subset of available information; this will allow for certain opinions to die out over time. Concretely, a reduction in diversity of sentiment displayed above may lead to the loss of arguments from negative opinions on controversial books, potentially crucial for historical or literary context or a nuanced understanding of a book's contributions.

Generative monoculture could even lead to security threats, depending on the application: software engineers across the globe relying on ChatGPT and CoPilot receiving similar code generations which do not reflect the true diversity of methods to solve a given problem may lead to similar code vulnerabilities across several large tech companies. Indeed, as we preview in Fig. 1 and show in detail in § 5, LLM output exhibits are less diverse than human-written solutions in the training data (e.g., by employing a narrower array of algorithms), which could lead to similarity across a wide range of code bases, leading in turn to repeated vulnerabilities (Perry et al., 2023; Pearce et al., 2022). However, in this case, LLM outputs not reflecting the full diversity of available coding examples can be positive: LLM outputs over-represent correct and efficient solutions. We present a nuanced picture of generative monoculture: while it can highlight the optimal portion of human-written data for some attributes, its pervasiveness across generation tasks and attributes *may* cause harm without careful intervention.

In this paper, we (1) define the concept of generative monoculture, and compare it to prior work around related topics (§ 2 and 7); (2) introduce a paradigm for measuring generative monoculture in LLMs (§ 3); (3) show experimental evidence for the prevalence of generative monoculture across a variety of application areas (book reviews and code) and (open-source and proprietary) LLMs, and provide some evidence for what may exacerbate generative monoculture, such as alignment tuning (Ouyang et al., 2022), (§ 5 and 6); and (4) show the (in)efficacy of several methods to abate generative monoculture: changing temperature, sampling, and prompting techniques (§ 5 and 6).

## 2    DEFINING GENERATIVE MONOCULTURE

We broadly characterize generative monoculture as a *distribution shift* from source data (i.e., human-written training data) to model generated data (i.e., model outputs) for a specific task, such as generating reviews for books or solutions to coding problems. This can be formalized using measures of statistical dispersion applied to various task-specific attributes.

**Definition 1** (Generative Monoculture). For a given task, let $\mathcal{P}_{\texttt{src}}$ denote the probability distribution of the source data, $\mathcal{P}_{\texttt{gen}}$ denote the probability distribution of the LLM-generated data, $h$ denote a function extracting attributes from data (such as sentiment, or algorithms used in code), and Dispersion$(\cdot)$ denote a dispersion metric (e.g., entropy). Then we define generative monoculture as the condition where $\mathcal{P}_{\texttt{gen}}$ is statistically narrower than $\mathcal{P}_{\texttt{src}}$, namely: Dispersion$(h(x)|x \sim \mathcal{P}_{\texttt{gen}}) <$ Dispersion$(h(x)|x \sim \mathcal{P}_{\texttt{src}})$.

Note, $\mathcal{P}_{\texttt{src}}/\mathcal{P}_{\texttt{gen}}$ can be the distribution of human-written/model-generated responses, for a given task, conditioned on one specific given prompt (which we refer to as the *conditional distribution*), or the distribution of human-written/model-generated responses for *any possible prompt* in a considered domain (which we call the *unconditional distribution*).

This phenomenon signifies a shift towards less varied outputs. We emphasize that the investigation of generative monoculture is intrinsically task-dependent, as the attributes of interest differ across tasks. In addition, as we often do not have access to the source distribution in practice, we approxi-

mate it using a source dataset ($D_{\texttt{src}}$), comprised of a subset of the training data of the LLMs. Similarly, we approximate the generated distribution through a dataset generated by the model ($D_{\texttt{gen}}$).

**Generative Monoculture, Human Preference, and Alignment:** Generative monoculture can cause LLMs to over-emphasize human-preferred areas of a distribution for a certain data attribute; this is often desired behavior. For example, as we demonstrate in § 6, generative monoculture can result in having a narrower distribution of code correctness or efficiency biased towards correct, fast, and low-memory code. We conjecture this is a consequence of alignment procedures such as reinforcement learning with human feedback (RLHF) Ouyang et al. (2022).

However, when a tendency stemming from human preference bleeds beyond its intended use—e.g., a preference for positive sentiment affecting outputs that need or should not be positive—these seemingly advantageous behaviors can prevent an equally important goal: *maintaining diversity of opinion and expression*. Further, along data attributes which do not have a clear "preferred area" of the distribution, generative monoculture can limit the scope of methods, topics, or ideas expressed.

## 3 MEASURING GENERATIVE MONOCULTURE

We outline a general approach to measuring generative monoculture in LLMs. In particular, following Definition 1, we outline steps to construct $D_{\texttt{src}}$ and $D_{\texttt{gen}}$ and compare their diversity through extracting data attributes and calculating dispersion metrics. We illustrate our approach in Fig. 2.

### 3.1 DATA CURATION

For a given task, we aim to create a source dataset that is likely to have been used in training the LLM we wish to investigate. Training data for most LLMs is a closely guarded secret. While recent work (Oren et al., 2023) describes how dataset contamination can be determined, such approaches are (a) riddled with false positives, and (b) computationally expensive. Thus, we often take an educated guess (based on dataset popularity and ease of use) in ascertaining if a given dataset is a likely training dataset candidate.

Formally, we define the source dataset as $D_{\texttt{src}} = \{q_i, \texttt{src}_i\}_{i \in [N]}$ where (a) $q_i$ is a problem instance within a task (e.g., name of a book for which a review has to be written), and (b) $\texttt{src}_i = \{\texttt{src}_i^j\}_{j \in [n_i]}$ is a set of $n_i$ human-written answers to the given prompt $q_i$ (e.g., a set of $n_i$ of book reviews for that particular book). In practice, we utilize existing datasets likely to be used during LLM training, and perform filtering and sub-sampling to obtain our $D_{\texttt{src}}$.

To create the model-generated dataset $D_{\texttt{gen}}$, for each sample $q_i$, we prompt the LLM ($\mathcal{M}$) we wish to evaluate, $m_i$ times to generate a set of responses, $\texttt{gen}_i = \{\texttt{gen}_i^j\}_{j \in [m_i]}$. Here, $\texttt{gen}_i^j \leftarrow \mathcal{M}_j(P_{\text{task}}(q_i), \texttt{kwargs})$ is the response obtained in the $j$-th call of $\mathcal{M}$, where (a) $P_{\text{task}}$ denotes the task-specific formatting prompt that wraps the sample $q_i$, and (b) $\texttt{kwargs}$ denotes the generation keyword arguments (e.g., temperature) that specify the sampling strategy. Across both $D_{\texttt{src}}$ and $D_{\texttt{gen}}$, we select or generate a large enough number of responses per $q_i$ to ensure variety.

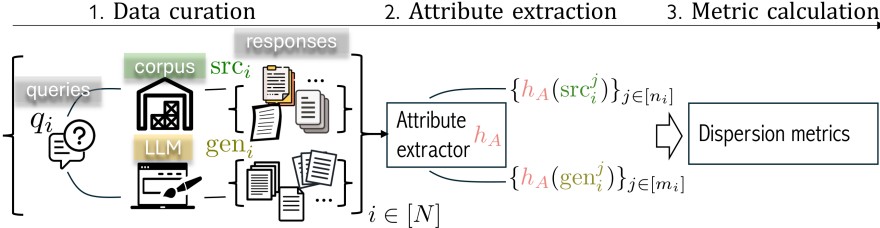

Figure 2: An overview of the procedure.

### 3.2 ATTRIBUTE EXTRACTION

For a given task, we identify and compile a list of attributes that are of interest from the perspective of preserving diversity. This is a subjective task, but we focus on metrics which target understanding the

content of LLM output (e.g., book review sentiment and topic; code time complexity and algorithms used), as opposed to more general metrics of output language quality such as saliency, fluency and coherence. More importantly, we need to ensure that extraction functions are efficient, accurate, and reproducible—we outline our tests to ensure these qualities in § 5 and the Appendices B and D. For example, care must be taken to use LLMs for attribute extraction, as they are known to be biased towards their own responses (Xu et al., 2024). For a given attribute $A$, extraction function $h_A$ takes a string `tgt` to obtain the attribute value $h_A(\text{tgt})$. Note that `tgt` can either be $\text{src}_i^j$ or $\text{gen}_i^j$. The extracted attribute can either be a continuous/categorical variable or of other more complicated types, depending on the nature of the attribute.

### 3.3 METRIC CALCULATION

As the last step, we compute metrics on the extracted attributes. Given a set of responses, dispersion metrics aim to capture their breadth or coverage of these attributes. We describe those used in this paper below and in Fig. 3.

**Dispersion Metrics.** We introduce dispersion metrics suited to different data types.

*A. Distribution of the mean.* For ordinal or continuous attributes, we calculate the mean over the conditional distribution—that is, we calculate the metrics over each $\text{src}_i$ or $\text{gen}_i$ (e.g. reviews for a given book), and show the distribution of this mean over all $q_i$ for a certain task (e.g. over all books). While the mean itself does not directly measure dispersion, the *distribution* of the mean values sheds light on dispersion: concentrated mean values indicates a smaller dispersion of the data. The advantage is that it not only describes dispersion, but also the qualitative tendency of the attribute (e.g. bias towards positive/negative sentiment), which cannot be captured otherwise.

*B. Entropy and standard deviation.* For categorical attributes, we measure dispersion using entropy over the conditional distribution. For continuous attributes, we use standard deviation over the conditional distribution to quantify the dispersion of values around the mean, providing the variability.

*C. Mean pairwise similarity.* For attributes that are not easily characterized as categorical or continuous, we adopt specific similarity metrics catered to the data type. We then calculate the pairwise similarity values for the conditional distribution—i.e. calculate the mean similarity value for all the pairs of elements within each $\text{src}_i/\text{gen}_i$, then show the distribution for all $N$ samples. We use:

1. *Mean pairwise Jaccard index*: Given two sets of categorical variables $A$ and $B$, the Jaccard index, $J(A, B) = |A \cap B|/|A \cup B|$, measures similarity between them. An example of such a set could be a set of several algorithms inferred from a piece of code. A higher mean Jaccard index indicates a higher overall similarity between the set, and consequently, lower dispersion.
2. *Mean pairwise cosine similarity*: Given two multi-dimensional embeddings $e_1$ and $e_2$ obtained via a sentence embedder (Reimers & Gurevych, 2019), we calculate their similarity via cosine similarity, i.e., $S_C(e_1, e_2) = \langle e_1, e_2 \rangle/\|e_1\|\|e_2\|$. A higher mean cosine similarity indicates a higher similarity and lower dispersion.
3. *Mean pairwise fingerprint similarity*: For tasks related to coding or computer programs, similarity is based on the overlap of selected hash values (or fingerprints) generated by Winnowing (Schleimer et al., 2003). We adopt an existing open-source tool COPYDETECT (Lingenfelter), which takes in a set of programs and returns the pairwise similarity scores for all programs in the set. We then calculate the mean value of these pairwise similarity scores as an indicator of the similarity for the set of programs. A higher mean fingerprint similarity indicateshigher structural and syntactical similarity of the code.

In addition to the dispersion metrics, we consider one other approach—visualizing the top modes of unconditional distributions for certain attributes, e.g., topics. This helps identify areas of emphasis in `src` and `gen` distributions, as well as the tendency of change across distributions.

## 4 MITIGATING GENERATIVE MONOCULTURE

To attempt to mitigate generative monoculture, we test four methods known to increase LLM output diversity: increasing the *temperature* $T$, top-$p$ parameter, setting a temperature decay, and changing prompts. More details are in Appendices A and C.6.

**Temperature** $T$**.** This determines the dispersion of the probability distribution over the next token: increasing the temperature leads to a more flat probability distribution and increases the likelihood of sampling from less probable tokens, resulting in more diverse generations.

**Top-**$p$**.** This controls the randomness of the generations by limiting the range of tokens considered. Specifically, it considers the smallest subset (consisting of the top probability tokens) whose cumulative probability exceeds the threshold $p$. A smaller $p$ encourages the model to sample from a more focused set of likely tokens.

**Decaying Temperature.** We choose the starting temperature $T = 10.0$ and follow a linear schedule for temperature decay, over the course of 50 time-steps (i.e., from the 1-st output token to the 50-th output token), with an ending temperature $T = 1.2$. The method is inspired by Carlini et al. (2021).

**Prompts.** Tuning the specific content and framing of the prompt can steer the model's output more effectively (Brown et al., 2020; Sclar et al., 2023) and significantly impact the diversity of the generated text. We use "role-playing" or impersonation (Salewski et al., 2024), which instructs the model to produce the output in the persona of a specific person, and expect it to induce more personalized and varied responses.

## 5 EXPERIMENTAL SETUP

In this section, we describe our experimental setup for measuring and mitigating generative monoculture for two tasks, namely, generating book reviews and code solutions. We provide details for datasets, LLMs used, and most notably, the data attributes and metrics considered. We open source our code at `https://github.com/GeMoLLM/GeMO`.

### 5.1 GENERATING BOOK REVIEWS

**Data Curation:** For $D_{\texttt{src}}$, we use the Goodreads dataset (Wan et al., 2019), which contains multiple books with several reviews each. We perform filtering and sampling to ensure reliable attribute extraction (see Appendix B.1), and craft a final dataset of $N = 742$ books with English titles, and $\forall i, n_i = 10$ reviews per book such that the review length is between 300 and 700 words.

To obtain $D_{\texttt{gen}}$, we used the following LLMs: (a) `Llama-2-13b` (Touvron et al., 2023) (henceforth referred to as `Llama-2`), (b) `Llama-2-13b-chat` (Touvron et al., 2023) (henceforth referred to as `Llama-2-chat`), (c) `Vicuna-13b-v1.5` (Chiang et al., 2023) (henceforth referred to as `Vicuna-13b`), (d) `GPT-3.5-turbo-instruct (0914)` (Ouyang et al., 2022) (henceforth referred to as `GPT-3.5`), and (e) `GPT-4-turbo (0125)` (Microsoft, 2024; Achiam et al., 2023) (henceforth referred to as `GPT-4`). We performed nucleus sampling (Holtzman et al., 2019) with various sampling parameters: (a) temperature $T \in \{0.5, 0.8, 1.0, 1.2, 1.5\}$, and (b) top-$p \in \{0.90, 0.95, 0.98, 1.00\}$. We also experimented with two candidates for $P_{\text{task}}$: prompt (1) "`Write a personalized review of the book titled {title}:`", and prompt (2) "`Write a book review for the book titled {title} as if you are {person}:`". Prompt (2) was chosen as LLMs are known to generate more diverse responses when instantiated with a persona (Salewski et al., 2024). We list the names of the 10 persons we considered in Appendix B.2. For comprehensiveness, we experimented with three more groups of prompts and report the results in Appendix C.7. For each combination of LLM, sampling parameter, and prompt, we independently sampled from the LLM 10 times to generate responses. We filtered out low-quality (generated) reviews by examining their perplexity (see Appendix C.1). This is to

| | Attribute | Data type | Level | Metric |
|---|---|---|---|---|
| Book review | sentiment | categorical | C | mean, entropy |
| | topic | categorical | C | entropy |
| | | | U | distribution visualization |
| | wording | categorical | U | count, entropy |
| Coding | correctness | categorical | C | mean |
| | efficiency (complexity) | categorical | C | entropy |
| | | | U | distribution visualization |
| | efficiency (runtime) | continuous | C | mean, standard deviation |
| | fingerprint | hash values | C | mean pairwise fingerprint similarity |
| | code summary (text) | embedding | C | mean pairwise cosine similarity |
| | code summary (categorical) | categorical | C | mean pairwise Jaccard index |

Figure 3: A summary of the scenarios, the attributes we consider, their data types, and the corresponding analysis levels as well as metrics. C and U stand for conditional and unconditional distributions.

ensure that the data used for analysis represents well-formed and coherent text, thereby improving the reliability of our findings. Thus, $\forall i, m_i \leq 10$.

**Attribute Extraction:** We want attributes that capture both the semantics and syntax of book reviews, representative of the key thematic and linguistic elements. Most importantly, while these attributes are not exhaustive, their extraction is reliable and efficient.

1. *Sentiment* indicates whether a review is positive (praising the book) or negative (criticizing the book). We employ a fine-tuned sentiment classifier (HuggingFace, b) as the attribute extractor which accepts text and returns a prediction in $\{0, 1\}$. This model has been downloaded $\sim 5.4$ million times, and reaches an accuracy of 91.3 % on the `dev` set of SST-2 (Socher et al., 2013).
2. *Topic* refers to the themes discussed in a review (Wallach, 2006; Alghamdi & Alfalqi, 2015). We leverage `BERTopic` (Grootendorst, 2022) pre-trained on Wikipedia (HuggingFace, a) which assigns one topic to each review out of a total of $\sim$2,000 topics.
3. *Word choice* captures the lexical diversity in a review. To quantify this, we produce a frequency table of the unique words (see Appendix B.3), and immediately have the number of unique words.

**Metric Calculation:** For sentiment, we calculate mean and entropy for the conditional distribution. For topic, we calculate entropy for the conditional distribution as well as visualize the unconditional distribution of topics across all reviews, focusing on the top 10 classes. Finally, for word choice, we calculate count and entropy of the unconditional distribution.

## 5.2 GENERATING CODE SOLUTIONS

**Data Curation:** For $D_{\text{src}}$, we chose the CodeContests dataset (Li et al., 2022), a competitive programming problem dataset where each problem comes with multiple correct and incorrect solutions. We limited the scope to a subset ($N = 100$) of level-A problems (easy problems) on Codeforces (CodeForces), and the language of the solutions to `python3`. More details in Appendix D.1. For each problem in the subset, we randomly sampled $\forall i, n_i = 20$ correct solutions from all of the $n_i^{\text{correct}}$ solutions for that problem.

To obtain $D_{\text{gen}}$, we use: (a) `GPT-4`, and (b) `Claude-3-Sonnet` (Anthropic, 2024). We did not use open-source LLMs, as these were not able to generate correct solutions for the problems we chose. More details are in Appendix E.4. We performed nucleus sampling (Holtzman et al., 2019) with various sampling parameters: (a) temperature $T \in \{0.5, 1.0\}$, and (b) top-$p \in \{0.9, 1.0\}$. We used only one candidate for $P_{\text{task}}$ i.e., "`Please read the below problem description and generate a python code to solve the problem {problem description} Please only generate code and nothing else.`" While we experimented with providing the LLM with a persona i.e., asking the LLM to pretend to be a "grandmaster in solving competitive programming problems", the resulting accuracy was lower (see Appendix E.2). For each combination of LLM, sampling parameter, and prompt, we produce $\forall i, m_i \geq 20$ generations such that at least 20 of the generated solutions were correct (details in Appendix D.2). We instantiated this by keep generating samples and measuring their correctness, until at some point all problems reached at least 20 correct solutions; we then stopped. This gave us $k = 100$ for `GPT-4` and $k = 200$ for `Claude-3`.

**Attribute Extraction:** We consider the following attributes which characterize different aspects of code. We rely on `GPT-3.5` for extracting some of the attributes; we manually verified the extracted attributes and confirmed their quality is high (see Appendix E.3).

1. *Correctness* refers to whether a piece of code correctly solves the given problem and passes all the test cases. We measure accuracy as the ratio of correct solutions among all solutions (details in Appendix D.3), to quantify the quality of human-/model-generated solutions.
2. *Efficiency* is crucial for scalability (Huang et al., 2024). This is measured through: asymptotic time/space complexity and runtime efficiency. We prompt `GPT-3.5` to infer the big $O$ time and space complexity (MacNeil et al., 2022), and execute the code on test cases to measure runtime and memory usage (see Appendix D.4).
3. *Fingerprint* provides insights into the structural and syntactical uniqueness of each code segment. As stated in Section 3.3, we use the COPYDETECT tool for this.

4. *Code Summary (textual)* explains the functionality of the code. Prior work has demonstrated the effectiveness of `GPT-3.5` in code understanding (Nam et al., 2024). Thus, we use it to produce text-based summaries, and a `description`, `functionality`, `algorithm`, and `data structure` (prompt for this task is in Appendix D.5). To compare the similarity for these text summaries, we produce their embeddings using the `all-MiniLM-L6-v2` model (Hugging-Face, c).

5. *Code Summary (categorical)* reflects the techniques employed in the code through categorical tags, as used on the Codeforces website. We prompt `GPT-3.5` to assign `tags` to a code segment by providing it a set of tags to choose from (prompt for this task is in Appendix D.5). We obtain one set per code segment. We similarly prompt `GPT-3.5` to choose from a list of `algorithms` and `data structures`.

**Metric Calculation:** For correctness, we calculate the mean value i.e., accuracy over the conditional distribution. For efficiency (asymptotic complexity), we calculate: (a) entropy for the conditional distribution, and (b) plot the histogram for the unconditional distribution. For runtime efficiency, we calculate mean and standard deviation for the conditional distribution. We measure the following over the conditional distribution: (a) fingerprints, where we calculate the mean pairwise fingerprint similarity; (b) code summary (textual), where we calculate the mean pairwise cosine similarity in their embedding space; and (c) code summary (categorical), where we calculate the mean pairwise Jaccard index.

## 6  RESULTS AND TAKEAWAYS

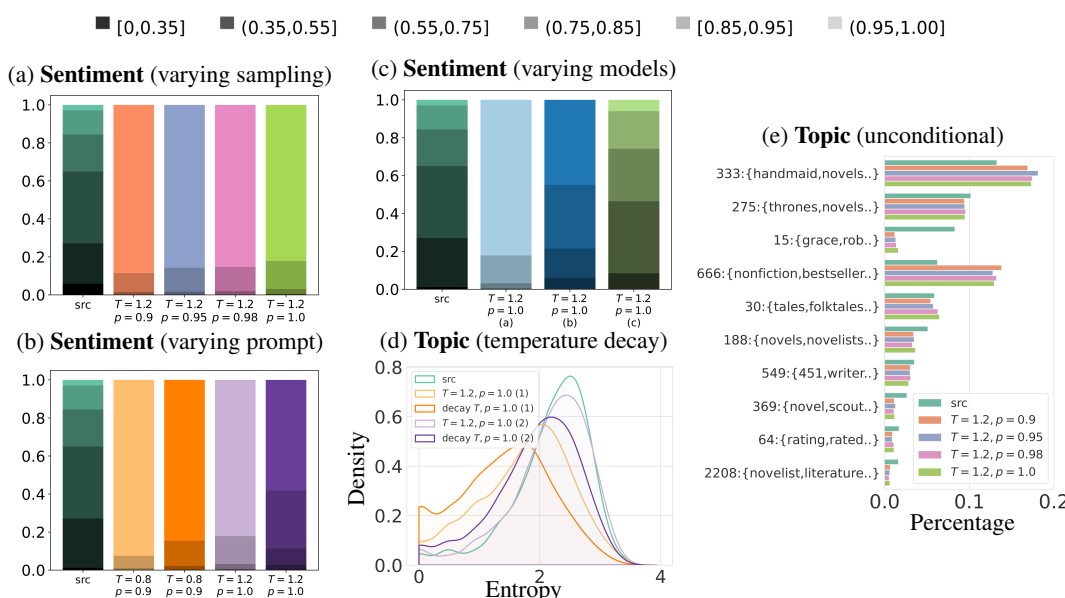

Figure 4: **(a-c)** stacked barplots for the mean sentiment scores under varying sampling parameters, prompts, and models. For these plots, in each bar, darker hues (bottom) represent lower scores while lighter one (top) denote higher scores. See the legend for the value range of each hue. In subfigure **(b)**, (1) and (2) refer to the two prompts as introduced in Section 5.1. In subfigure **(c)**, (a-c) refer to `Llama-2-chat`, `Vicuna-13b`, and `Llama-2`. **(d)** kernel density estimation (KDE) for the entropy values calculated on the conditional distribution of the topics. **(e)** unconditional topic distribution for top-10 topics. For all the subfigures, we mark the sampling parameters in them; unless marked with (1-2) or (a-c), the results are obtained on `Llama-2-chat` under prompt (1). These subfigures show that the model-generated reviews are overwhelmingly positive and cover a narrower range of the topics per book; moreover, there is distinctive under- and over-representation of the topics covered overall.

**Guide:** We present our results on measuring, and attempting to mitigate, generative monoculture in Fig. 4 and 5. We display results mainly in three formats: (a) stacked bar charts, where different hues correspond to different value ranges as indicated in the legend; (b) histograms (or grouped bar charts), to reflect the probability mass of a categorical variable; and (c) kernel density estimation

(KDE) plots, to reflect the estimated probability density of a continuous variable. We note that, in the code results, for all plots except that evaluating accuracy, we restrict to correct solutions.

**Takeaway 1: Monoculture Exists and is Severe, Within and Across LLMs.** As shown in Fig. 4 and 5, there exists significant narrowing from the source to generation distribution in all attributes considered for both scenarios, book reviews and coding.

Notably, for book review, *proprietary OpenAI LLMs (GPT-3.5 and GPT-4) demonstrate even more severe monoculture compared with the open-source Llama family LLMs* (see Fig. 9 in Appendix C.5 for more details). Particularly, for both GPT-3.5 and GPT-4, 100% of the samples have average positivity falling in (0.95,1.00] under prompt (1), and 98.9% and 97.6% under prompt (2). For coding, similar reductions in diversity can be seen: in Fig. 5(e), we see increased similarity in natural language descriptions of LLM-generated code solutions, and Fig. 5(f) shows the Jaccard similarity of the generated solutions in terms of the inferred algorithms, with the majority of problems displaying high similarity across generated solutions. Of particular interest, the plagiarism scores of the LLM-generated code are extremely high (Fig. 5(b)), compared to the source solutions which achieve an utterly zero plagiarism score for *all* the problems. We examine a few pairs of examples and their plagiarism scores in Appendix E.1.

**Takeaway 2: LLMs tend to produce Human-favorable Generations.** Our results show that LLMs tend to over-represent parts of the attribute distribution that are preferred by humans: humans largely prefer text with positive sentiment (Dodds et al., 2015; Augustine et al., 2011; Boucher & Osgood, 1969) as well as correct and efficient code, and researchers have specifically infused these preferences into LLM assistants through preference tuning (e.g. RLHF) (Ouyang et al., 2022; Bai et al., 2022; Roziere et al., 2023). Fig. 4(a) and Appendix C.5 show that LLMs produce overwhelmingly positive generations. Fig. 5, as well as Fig. 22 and 23 in the Appendix reveal that LLM-generated code segments (a) are over $2\times$ more accurate than the average human solutions, (b) enjoy an overall lower asymptotic time and space complexity, and (c) use less runtime and memory during execution. This may just be the intended consequence of RLHF, which explicitly optimizes the LLM towards producing human-favored responses in its objective, as guided by a reward model trained on human preferences.

However, as our results show, this implies a loss of diversity guided by human preferences, which, if only naively understood and enforced, could lead to unwanted consequences if going unnoticed. One example of the unintended artifacts is the under- and over-represented topics (Fig. 4(e)); the topic group 15 which contains keywords "rob" and "kill" etc. is significantly under-represented, likely a consequence of RLHF alignment tuning.

**Takeaway 3: RLHF Hurts Diversity the Most.** Llama-2-chat is obtained via performing RLHF tuning (Ouyang et al., 2022) on the pre-trained (PT) Llama-2. Similarly, Vicuna-13b is obtained via supervised fine-tuning (SFT) on the PT Llama-2 (Chiang et al., 2023). Comparisons on these LLMs (see Fig. 4(c), as well as Fig. 8 in Appendix C.4) show that the PT LLM-generated reviews are much more similar to the source. The PT LLM Llama-2 has 5.9% of samples with average sentiment values falling in the range of (0.95,1.00], which is much closer to the source percentage of 2.8% than 44.7% for Vicuna-13b and 82.1% for Llama-2-chat. Vicuna-13b also shows better diversity than Llama-2-chat—this is consistent with findings suggesting RLHF reduces output diversity compared with SFT (albeit with different metrics) (Kirk et al., 2024).

**Takeaway 4: Naive Mitigations are Insufficient.** Changing the sampling parameter (increasing $T$ and $p$) and using a more diversity-inducing prompt (e.g., prompt (2) for book reviews) can reduce the gap (see Fig. 4(a-b) and Fig. 5). For example, using prompt (2) reduces the percentage of the most positive range from 82.1% to 58.1% in Fig. 4(b) for $T = 1.2, p = 1.0$. However, the gap is still large. More results in Appendix (Figures 13 and 14, and Appendix E) show similar conclusions.

We attempted two other strategies: (a) picking a higher temperature, and (b) leveraging a decaying temperature scheme (see § 4). Results in Appendix C.2 show that the gap still remains big even at such high randomness. Furthermore, for larger $T$, we notice a significant degradation of the generation quality as a result of the increased randomness. In Table 1, we present the average fraction of valid generations for Llama-2-chat and Vicuna-13b under various sampling parameters. The table shows that the valid number of generations rapidly drops as the randomness increases, particularly at $T = 1.5$; the implication is that such a high randomness setting cannot be adopted for practical use.

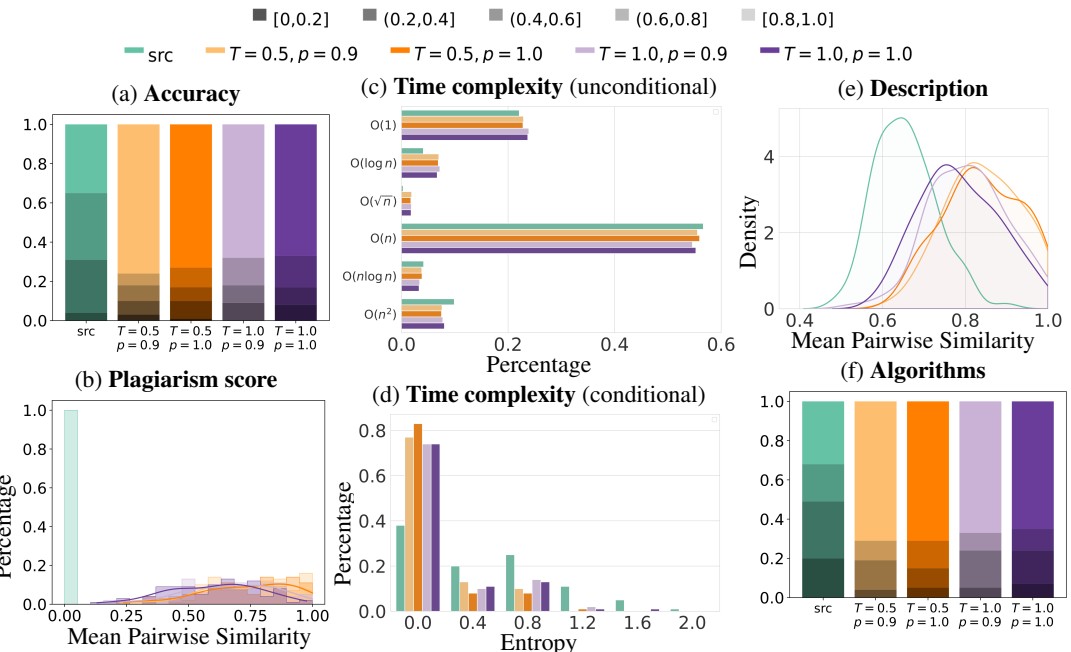

Figure 5: (**Left**) (**a**) stacked barplot for accuracy and (**b**) probability mass along with KDE for plagiarism scores. (**Middle**) Time complexity: (**c**) histogram of the (unconditional) distribution of the asymptotic complexity and (**d**) probability mass for the (conditional) distribution of entropy values. (**Right**) Selected code summary: (**e**) KDE plot for the mean pairwise cosine similarity scores for "description" as natural language and (**f**) stacked barplot for the mean pairwise Jaccard scores for "algorithms" as categorical values. Overall, the model-generated solutions are more accurate and efficient, display higher description similarity to each other, and cover a narrower span of algorithms. (More results in Appendix E.5.)

Table 1: **The average fraction of valid generations (out of a total of 10)** for two models under various sampling parameters (temperature, top-$p$, and prompt–denoted (1) and (2)). We regard a "valid" generation as text of perplexity value $\leq 20$—as a support, we present high perplexity samples in Appendix C.1. We observe that the number drops as the randomness increases (along the increase of both $T$ and top-$p$ values). As a reference, GPT-4 achieves an average ratio of 1.000 at $T = 1.2$ and $p = 1.0$.

| LLama-2 -13b-chat | $p = 0.90$ | | $p = 0.95$ | | $p = 0.98$ | | $p = 1.00$ | | Vicuna -13b | $p = 0.90$ | | $p = 0.95$ | | $p = 0.98$ | | $p = 1.00$ | |
|---|---|---|---|---|---|---|---|---|---|---|---|---|---|---|---|---|---|
| | (1) | (2) | (1) | (2) | (1) | (2) | (1) | (2) | | (1) | (2) | (1) | (2) | (1) | (2) | (1) | (2) |
| $T = 0.5$ | 1.000 | 1.000 | 1.000 | 1.000 | 1.000 | 1.000 | 1.000 | 1.000 | $T = 0.5$ | 0.987 | 0.992 | 0.987 | 0.990 | 0.984 | 0.989 | 0.984 | 0.988 |
| $T = 1.0$ | 1.000 | 1.000 | 0.999 | 1.000 | 0.999 | 1.000 | 0.998 | 0.999 | $T = 1.0$ | 0.961 | 0.962 | 0.951 | 0.958 | 0.942 | 0.953 | 0.935 | 0.947 |
| $T = 1.5$ | 0.994 | 0.988 | 0.930 | **0.883** | 0.743 | 0.649 | 0.512 | 0.394 | $T = 1.5$ | **0.871** | 0.903 | **0.835** | 0.876 | 0.766 | 0.840 | 0.680 | 0.767 |

Though prior work showed promise for decaying temperature to encourage diversity while maintaining quality (Carlini et al., 2021), this too failed to achieve higher diversity (see Fig. 4(d) and Appendix C.6).

# 7 RELATED WORK

**Diversity and LLMs.** Santurkar et al. (2023) demonstrate that LLMs do not give representative opinions to polling questions when compared to the general U.S. population. Our work focuses on the narrowing of diversity in LLM output from its human-written training data—while Santukar *et al.* demonstrate a narrowing in diversity from actual human survey respondents (and not training data). Additionally, our work proposes a general framework for measuring monoculture. Padmakumar & He (2023) demonstrate that using LLM assistance can lead to reduced diversity in human-written argumentative essays when compared to essays written without LLM assistance. While they mention that this is partially because the models themselves do not produce diverse output, they do not focus on the narrowing of diversity from LLM training data to LLM-generated data. Finally, Zhang et al. (2024) propose an approach to fine-tune LLMs to output desired target distri-

butions, and Sorensen et al. (2024) outline an alignment framework to emphasize *pluralism* to work towards creating models which express a variety of opinions and perspectives. While these are certainly related to our work, generative monoculture as a phenomenon extends beyond differences in opinion, and expresses the narrowing of any number of task-specific attributes, from code correctness to topics covered to many others. One common thread across many of these works, which our work adds to, is that *current alignment practices—namely RLHF— harms output diversity*.

**Other Notions of Monoculture.** Our notion of generative monoculture relates to, but differs from, other notions of monoculture in the AI literature. For example, algorithmic monoculture (Kleinberg & Raghavan, 2021) and outcome homogeneity (Bommasani et al., 2022) describe the societal state where many decision-making actors rely on the same underlying algorithms to generate (classification or ranking) predictions, from the perspective of decision-making actors and individuals subject to those decisions respectively. These works show that algorithmic monoculture is sub-optimal for both decision-making actors (due to correlated failures across models) and for those subject to model decisions, as repeated outcomes across models leave little room for algorithmic recourse. In contrast, generative monoculture focuses on documenting the phenomenon of individual LLMs narrowing the diversity of their output in relation to their source data—for example, only returning positive book reviews about a controversial book. We do, however, document in this work that generative monoculture exists to similar extents and in similar directions across a variety of available LLMs, (e.g., Llama, Vicuna, ChatGPT-4) leaving open the possibility of concerns brought up by Kleinberg & Raghavan (2021), but in a generative context.

**Connections to Model Collapse:** We evaluate models trained on human-curated data, whereas model collapse evaluates models trained iteratively on synthetic data (either fully, or mixed with human data). In settings of model collapse where the model is trained only on synthetic data (Shumailov et al., 2023; Taori & Hashimoto, 2023), it is understandable that the generation quality is low. In contrast, our work shows that the generation quality is good (e.g., model generates correct coding solutions), but the "diversity" in generations is low. Unlike model collapse which converges to the mean of the distribution (Shumailov et al., 2023), our observation is there's an emphasis on a specific part of the distribution which is not necessarily the mean. In this way, the work is tangentially related to collapse, but is not a special case of it (as the collapse phenomenon necessitates the distribution to match the mean with many rounds).

## 8 CONCLUSION AND LIMITATIONS

In this work, we introduce the concept of generative monoculture, a phenomenon where LLMs narrow the diversity of their output relative to their source data for a given task. We experimentally demonstrate its prevalence across text and code generation tasks, and show the difficulty in mitigating the behavior. Our work has limitations: first, we did not analyze the full training set of the LLMs we study due to time and compute restrictions, as the corpora are large and often proprietary. Further, as we note in § 3, measuring monoculture is difficult as selecting attributes is subjective, and the attribute extraction process is sensitive to the reliability of extraction techniques. (We verify our own attribute extraction techniques in the appendix). Further, while generative monoculture itself can have unfair consequences by enforcing the suppression of minority opinions, mitigating monoculture without extreme care could lead to the proliferation of harmful ideas or even toxicity by allowing for representation of the entire distribution of source text. We look forward to future work mitigating monoculture while maintaining low levels of toxicity and other dangerous behavior.

## ACKNOWLEDGEMENTS

This research was supported in part by the Accelerating Foundation Models Research grant from Microsoft.

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

CONTENTS

## APPENDIX

We open source our code at `https://github.com/GeMoLLM/GeMO`.

## A  SAMPLING PARAMETERS

**1. Temperature:** Concretely, the temperature parameter $T$ determines the "dispersion" of the probability distribution over the next token. Mathematically, the probability of a token $w$ being generated is given by the softmax function: $P(w|x) = \frac{\exp(s(w|x)/T)}{\sum_{w'} \exp(s(w'|x)/T)}$ where $s(w|x)$ is the unnormalized log-probability of the token $w$ given the context $x$. Increasing the temperature leads to a more flat probability distribution and increases the likelihood of sampling from less probable tokens, resulting in more diverse generations. This can also be understood from the perspective of entropy of the next token, $H(W) = -\sum_w P(w|x) \log P(w|x)$, where it can be seen that the increase of entropy directly follows.

**2. Top-$p$:** The top-$p$ parameter Holtzman et al. (2019) ($p \in (0, 1]$) controls the randomness of the generations by limiting the range of tokens considered. Specifically, it considers the smallest subset (consisting of the top probability tokens) whose cumulative probability exceeds the threshold $p$. A smaller $p$ encourages the model to sample from a more focused set of likely tokens, while a larger $p$ allows sampling from a broader range and thus increases randomness ($p = 1$ basically means no restriction on the vocabulary).

Different platforms have different default values for $T$ and top-$p$. `GPT-3.5-turbo-instruct (0914)` web version adopts $T = 0.8$ Community (2023). The default values in OpenAI APIs are $T = 1.0$ and $p = 1.0$ OpenAI (2024).

**3. Generation Length:** This parameter (`max_new_tokens`) dictates at most how many tokens the model should generate before it stops. We used 500 for book review generation and 2048 for code generation.

# B ADDITIONAL DETAILS: BOOK REVIEWS

## B.1 CONSTRUCTION OF THE SOURCE DATASET

To ensure we picked popular books that the LLMs know about, we filtered the books according to the number of reviews they have. Constraining the number of reviews to be between 1,000 and 2,500, we obtained 750 books. We further filtered out books with non-English titles; we conducted this filtering because some downstream LLMs (e.g., sentiment classifiers) are not multilingual, and as a result can not analyze the generated non-English reviews. To ensure high quality of the reviews, we further filtered the review by length, constraining the length of each review to be between 300 and 700 words. After this step of filtering, we sampled 10 reviews per book. Eventually, we obtained a dataset of 742 books, where each book comes with 10 reviews.

## B.2 NAMES OF THE CELEBRITIES USED IN PROMPT 2

We prompted `GPT-4` to provide a list of celebrities suitable for writing diverse book reviews. The list of names are as follows: Trevor Noah, Janelle Monáe, Yuval Noah Harari, Serena Williams, Reshma Saujani, Neil deGrasse Tyson, Margaret Atwood, David Attenborough, Malala Yousafzai, Jordan Peele.

## B.3 TEXT PROCESSING FOR ANALYZING WORDING CHOICE

We first concatenated all documents and converted them to lowercase to standardize the text. We then expanded contractions (e.g., "don't" to "do not"). We then tokenize the text with `word_tokenize()` from the NLTK library Bird & Loper (2004), and removed the punctuation. We continued by filtering out non-alphabetic tokens to focus solely on words. Lastly, we lemmatized tokens via `WordNetLemmatizer()` to their base forms, aiding in consistent frequency analysis. These steps are essential for minimizing textual noise and ensuring the reliability of our word frequency assessments.

# C ADDITIONAL RESULTS: BOOK REVIEWS

## C.1 FILTERED-OUT REVIEWS

We established a perplexity threshold of 20 to filter out low-quality reviews. To validate our choice, we randomly sampled reviews with perplexity scores at different intervals and manually inspected them. For clarity, instead of presenting the entire review text, we selectively extracted chunks that exemplified the low-quality nature of the sampled reviews.

> **Perplexity in** $(20, 25]$: This page-turning, and fast-paced gripping story makes excellent use both tropes and novel ideas by employing both like, for instance, a dystopian setting which uses an exaggeratory ruminating setting so typically found in series belonging to this genre, only that here the author's unique skill at weaving such plot-lines together in a novel that feels fresh more like Harry Potter but with an 800word length page each chapter instead of JUST 200word paragraph at the end of an excrutiatingly slow chapter.

> **Perplexity in** $(25, 30]$: That and artist Adrian Alphona bringing the world-sprawling amazing action and gorgeous characters from beautifully rendered backgrounds in color by vivid and dazzling color is what makes the graphic novel in itself even more delightful. With each color panel bringing an explosion of color onto every page no panel is left the same no character or page lacking the same amount of energy vivid and gripping from the moment I turned on. the illustrations are bright beautiful detailed it truly does it live upto its tag line and then some a hero like no other a book like no other definitely worth the dive.

> **Perplexity in** $(30, 35]$: One would wonder if she lived some of this amazing journey to unhidden holes in the ground or lived the stories and characters by the holes they created one will see how amazingly her fantastic works has a way t connect the reader to a place which feels so real and fascinating! Holes by Loues Erdrich has a beautifully constructed a magical and a world which transports and takes readers like on a journey thrifting us so many magical and real places which we might not have an opportunity if it wasn't for the eyes and minds eye of Louise Erdrech .

> **Perplexity in** $(35, 40]$: It made me reflect for a deeper appreciation on both the power of memory in creating some of literature and art's most significant works or cultural touchstone masterpieces such as those listed that our lead female artist was known to adore and how that and the themes that author – a most wonderful story teller by the way, by way of characters that i wanted or did want to spend more time with and learn the most in depths exploration of, and not all can reach it. And so a deeper appreciation for the art of storytelling from great narratives by great narrators? The Little Paris Bookp shop does achieve an astounding success in that regard, I am thrilled I'd like to shout it out to any and All in ear shot about it!

## C.2 GENERATION RESULTS AT HIGHER RANDOMNESS

We experimented with even higher randomness at $T = 1.5$ motivated by the observation that increasing the randomness does help to increase the diversity. However, as we show in Fig. 6, even at high randomness, there is still a huge gap between the source and the generations.

Moreover, we notice a significant degradation of the generation quality as a result of the increased randomness. We present in Table 1 the average number of valid generations for two models under various sampling parameters. The table shows that the valid number of generations rapidly drops as the randomness increases, particularly at $T = 1.5$; the implication is that such a high randomness setting basically cannot be adopted for practical use.

## C.3 TOPIC SHIFTS

We present in Fig. 7 the results of the unconditional topic distribution. Across all three settings for comparison, we observe highly similar distribution shift signified by the over- and under-representation of certain topic groups. Concretely, the topic group 15 with the keywords "grace, rob, nick, anna, house, discovers, realizes, killed, confronts, killer" are under-represented, potentially because certain words like "rob" and "kill" are *eliminated* from the output as a result of RLHF. On the other hand, the topic group 333 with the keywords "handmaid, novels, novel, writers, tale, book, books, literary, novellas, synopsis" and the topic group 666 with the keywords "nonfiction, bestseller, novelist, autobiography, novels, memoir, author, memoirs, paperback, novel" are over-represented. The over-representation of topic group 333 in `Vicuna-13b` is much more severe than `Llama-2-chat`, as can be seen from the 3rd subfigure; this could be because a higher exposure to relevant materials during its fine-tuning.

## C.4 PRE-TRAINED MODEL

The aligned model is obtained from performing supervised fine-tuning followed by RLHF alignment tuning on the basis of the pre-trained model Touvron et al. (2023). We compare the performance of the aligned model llama-2-13b-chat with the pre-trained model llama-2-13b and present the results in Fig. 8.

From the 1st and 2nd subfigure, we observe that the pre-trained model is much more diverse than the aligned model; the pre-trained model is even very close to the source data (see the dark purple result tagged as $T = 1.2, p = 1.0$ (c)).

The 3rd subfigure delivers two messages. First of all, there still exists divergence from the pre-trained model and the source in terms of the covered topics. We attribute this divergence we observe between the pre-trained model and the source data to the difference between the source data we use (the Goodreads dataset) and the ground-truth training data (a much broader corpus which we have no information about). We regard the Goodreads dataset as a proxy of the ground-truth, but intrinsically Goodreads is smaller and does not fully accurately represent the groundtruth training distribution, especially in nuanced attributes like the unconditional topic distribution measured on all samples. Second, the direction and trend of changes in the aligned model is not completely the

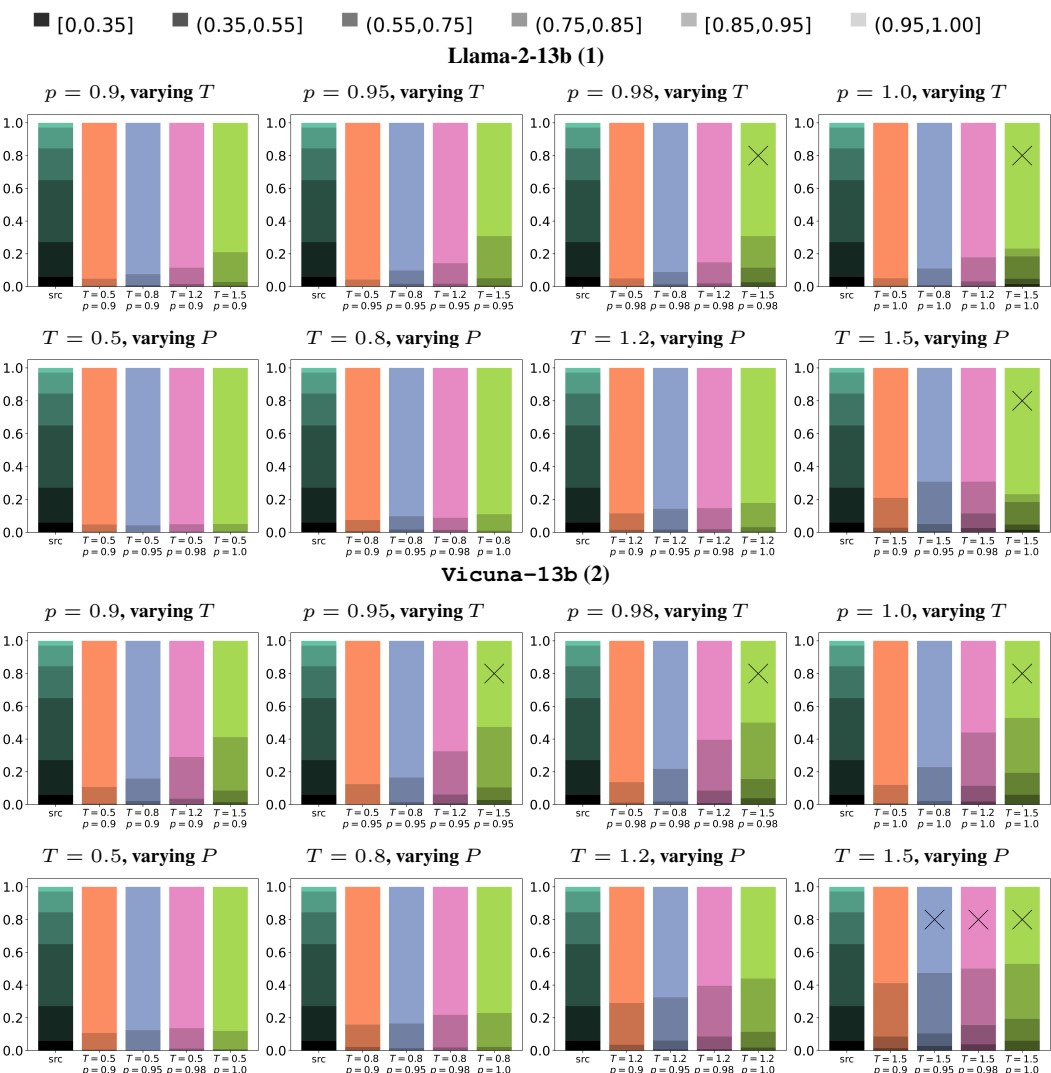

Figure 6: **Mean sentiment scores** across different models (`Llama-2-chat`, `Vicuna-13b`) and prompts ((1) and (2)), and varying sampling parameters (temperature $T$ and top-$p$). We do observe an increase of diversity along the increase of generation randomness (i.e., increasing $T$ and $p$), but the gap remains high between the source and even the highest generation randomness setting. Moreover, we put a black cross $\times$ on the top of the setting with a concerning low number of valid number of generations (see Table 1 for a concrete explanation); they mostly only occur for the few highest randomness settings.

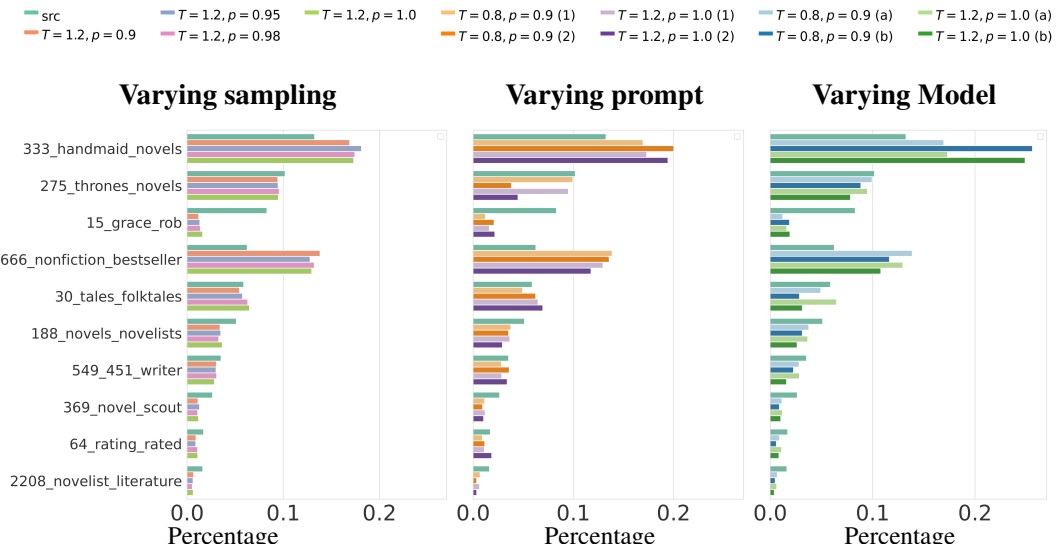

Figure 7: **Grouped bar charts of the unconditional distribution of the topic** (i.e., the distribution of the topic over all the samples). (**Left**): generated by `Llama-2-chat` at different generation kwargs, where we fixed $T = 1.2$ and varied $p \in \{0.90, 0.95, 0.98, 1.00\}$. The prompt used is (1) as introduced in Section 5.1. (**Middle**): generated by `Llama-2-chat` under different prompts (1) and (2). (**Right**): generated by two models (a) `Llama-2-chat` and (b) `Vicuna-13b`, using the prompt (1). Across all three subfigures, we observe highly similar distribution shift signified by the over- and under-representation of certain topics.

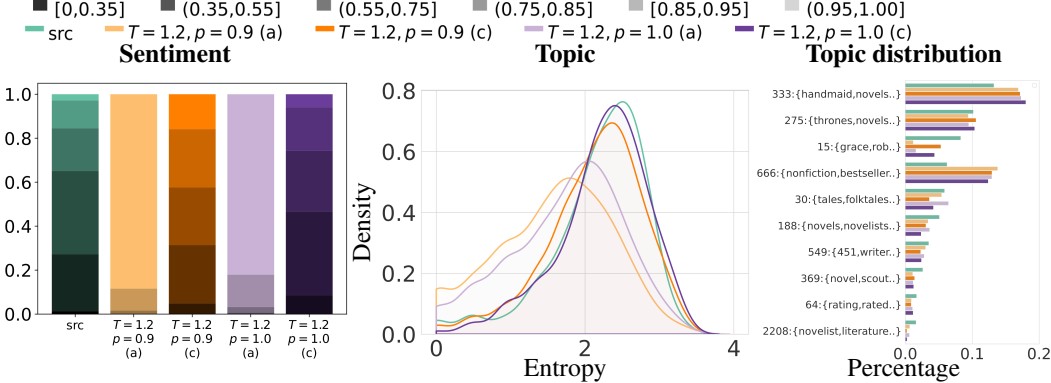

Figure 8: **Comparison between (a) the *aligned* model `Llama-2-chat` and (c) the *pre-trained* model `Llama-2`**, under prompt (1), temperature $T = 1.2$, and varying top-$p$ parameters. We present a subset of the results (sentiment, topic, and topic distribution), corresponding to Figure 7. The figures show that the pre-trained model enjoys much better diversity than the aligned model, and is much closer to the source.

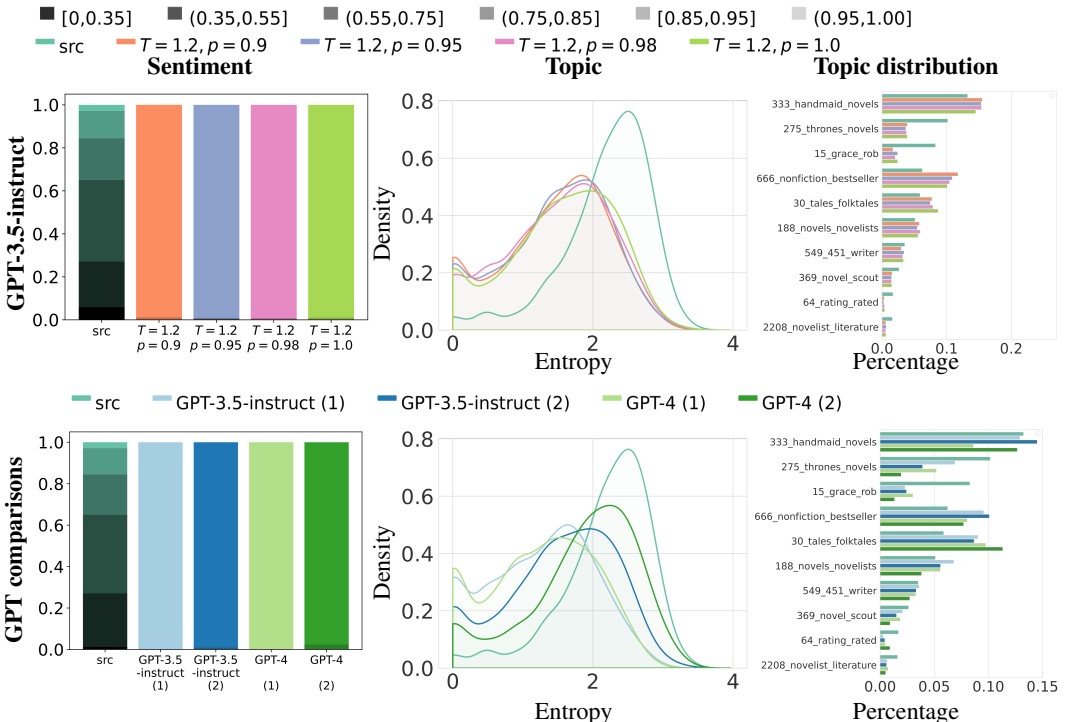

Figure 9: **Plots for OpenAI models**. **(Upper row)**: `GPT-3.5` under prompt (1), $T = 1.2$, and varing $p$. **(Lower row)**: comparing `GPT-3.5` and `GPT-4` at $T = 1.2, p = 1.0$ and the two propmts (1) and (2). We present a subset of the results (sentiment, topic, and topic distribution), corresponding to Figure 7. The figures show that both `GPT-3.5` and `GPT-4` display severe GeMo, which cannot be effectively mitigated via varying the sampling parameters or using a diversity-inducing prompt.

same as the aligned models, e.g., the topic group 15 containing "rob" and "killer" is not inhibited as much as in the aligned models, potentially due to RLHF.

## C.5 RESULTS OF OPENAI MODELS

We repeat the same set of experiments on the proprietary OpenAI models GPT-3.5-turbo-instruct (0914) and GPT-4-turbo (0125). We present the results in Fig. 9.

The sentiment results (in column 1) suggest that both models are overwhelmingly and invariably positive in their reviews, across various sampling parameters and prompts.

The results of the conditional topic distribution (in column 2) reveal similar conclusions—there is a significant deviation between the generated reviews and the source reviews across varying sampling parameters (see upper low) and for both GPT-3.5-instruct and GPT-4 (see lower row). Nevertheless, using a diversity-inducing prompt, i.e., prompt (2) does lead to increased diversity (see lower row).

The results of the unconditional topic distribution (in column 3) also demonstrate the distribution shift, though in a slightly different way than observed for llama family models (see presented in Appendix C.3). Concretely, the topic group 333 does not experience a significantly over-representation, while the topic group 30 does. As discussed previously, we attribute the difference in the nuanced topic distribution mainly to the difference in the training data.

## C.6 MITIGATION VIA TEMPERATURE DECAY

We describe the decaying temperature scheme. Concretely, we choose the starting temperature $T = 10.0$ and follow a linear schedule for temperature decay, over the course of 50 timesteps (i.e.,

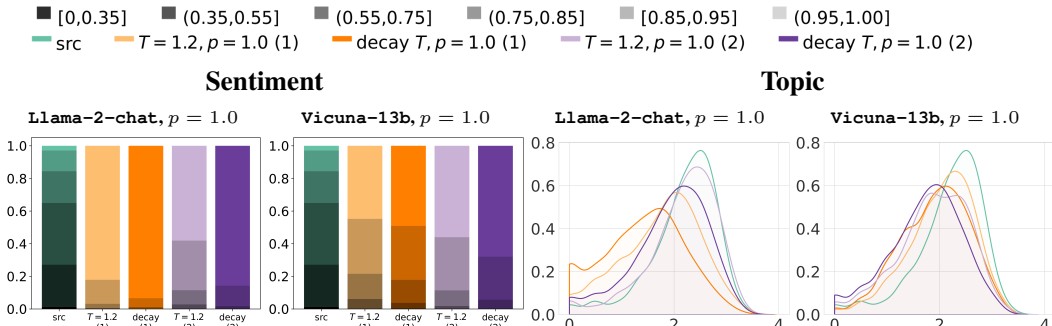

Figure 10: **Effect of temperature decay during decoding time** for two models `Llama-2-chat` and `Vicuna-13b`, with top-$p$ parameter 1.0 and two prompts (1) and (2). We compare the fixed temperature ($T = 1.2$) with the decaying temperature (decay $T$) and we present the results for sentiment and topic. The figures reveal the ineffectiveness of the temperature decay method.

from the 1-st output token to the 50-th output token), with an ending temperature $T = 1.2$. The method is inspired by Carlini et al. Carlini et al. (2021) which reported this scheme as one sampling method that induce diversity and high quality output[1].

We present the results in Fig. 10. For both the sentiment and the topic, we see that the fixed temperature scheme achieves a higher diversity compared to the decaying temperature scheme.

## C.7 EVALUATION ON ADDITIONAL PROMPTS

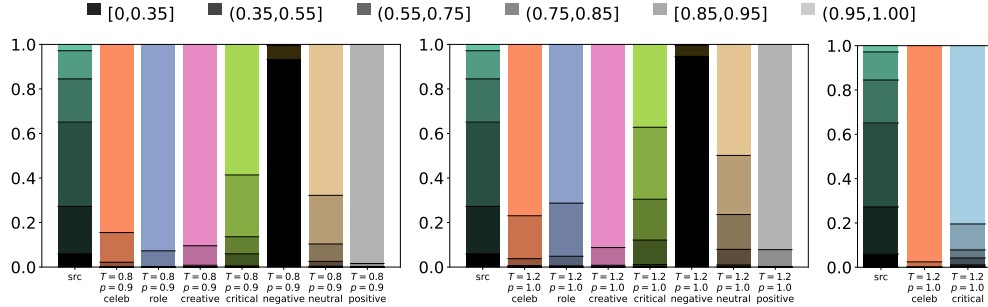

Figure 11: **Stacked barplots for the mean sentiment scores.** Left and middle for Llama-2-13b-chat and right for GPT-4. The label "celeb" corresponds to Prompt (2) in the main submission; the prompts for other labels can be found below. The all black bar for the label "negative" means that all the generations for this prompt achieve extremely low scores (refer to the hue legend).

Aside from prompts (1) and (2) introduced in the main paper in § 5.1, we additionally experimented with three groups of prompts for comprehensiveness.

**Group 1** [specifying detailed roles]:

- Prompt (role): "`Write a book review for the book titled {title} as if you are a {role}:`", where role ∈ {"teenage fantasy enthusiast", "critical literature professor", "romance novel lover", "tech-savvy sci-fi geek", "history buff", "casual weekend reader", "book club moderator", "non-fiction aficionado", "poetry appreciator", "mystery thriller addict"}

- Prompt (creative): "`Write a book review for the book titled {title}, from the viewpoint of different personas, such as 'aspiring`

---

[1]We refer to `https://github.com/shreyansh26/Extracting-Training-Data-from-Large-Langauge-Models/blob/main/extraction_temperature_decay.py` for an implementation of the temperature decay sampling scheme for HuggingFace models.

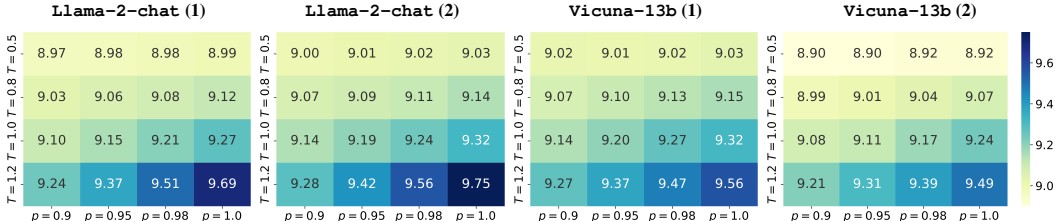

Figure 12: **Heatmap of the entropy of the (unconditional) distribution of the *word choice* in model-generated reviews**. We present results of two models (`Llama-2-chat` and `Vicuna-13b`) and two prompts ((1) and (2)). As a reference, the entropy of the source is 9.99, higher than all the entropy values of model generations.

```
writer', 'history enthusiast', 'teenage sci-fi fan', or
'career-focused parent', etc.  Be creative!"
```

**Group 2** [Steering attributes such as sentiment]:

- Prompt (negative): "`Write a negative review of the book titled {title}:`"
- Prompt (neutral): "`Write a neutral review of the book titled {title}:`"
- Prompt (positive): "`Write a positive review of the book titled {title}:`"

**Group 3** [Other variations]:

- Prompt (critical): "`Write a critical review of the book titled {title}:`"

We conducted the additional experiments on a) `Llama-2-13b-chat` and b) `GPT-4`. For the former, we evaluated two sampling parameters ($T = 0.8, p = 0.9$ for relatively low randomness, and $T = 1.2, p = 1.0$ for high randomness); for `GPT-4`, we considered only the high randomness parameter and only the `prompt (critical)` due to the long runtime. For each combination of model, prompt, and sampling parameter, we generated $n = 10$ reviews.

We present the results in Figure 11 and discuss the takeaways below.

1. `Prompt (critical)` achieves the best diversity among all. However, there is still a significant gap from the `src`, which holds true for both models.

2. `Prompt (role)` achieves a similar level of diversity compared to `Prompt (person)`, showing that specifying a diverse set of *detailed roles* do not bring improvement over specifying a diverse set of *celebrities*. Giving the LLM extra freedom to explore using different roles (i.e., `Prompt (creative)`), however, leads to even worse diversity.

3. Using sentiment words (i.e., negative, neutral, positive) can indeed steer the sentiment of the generations very effectively. Yet we clarify that the model being a good instruction follower does not directly translate into it preserving diversity, without a good instruction giver. Moreover, as per our definition, we care about diversity of multiple attributes. Explicitly steering one attribute is insufficient for achieving diversity across numerous attributes. More generally speaking, the issue resides in—what we can control/steer is a limited set of knowns, but what we hope to gain is w.r.t. a more broader set of knowns and unknowns.

## C.8   MORE RESULTS

For the attribute *word choice*, we present the count of unique words in Table 2 and the entropy for the word distribution in Fig. 12, showing that model-generated reviews use a narrower vocabulary and is less diverse than the human-written reviews.

We present more results for various attributes and metrics in Fig. 13 and 14 where we vary several factors for detailed comparisons. Overall, the results suggest the prevalence and severity of generative monoculture, as well as the ineffectiveness of naive mitigations.

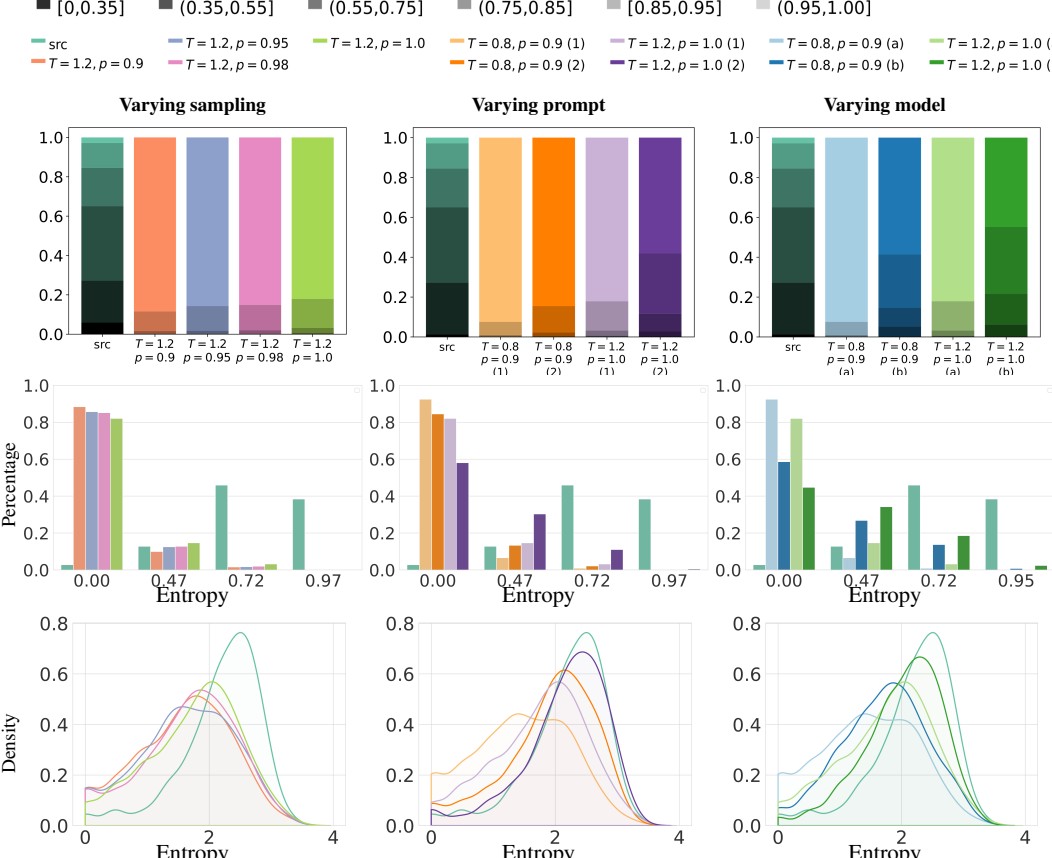

Figure 13: **(Row 1)**: Stacked bar charts of the mean sentiment scores. In each bar, darker hues (bottom) correspond to the lower scores while the lighter hues (upper) are the higher scores. See the legend for the detailed value range of each hue. **(Row 2)**: Histograms of the entropy of the sentiment scores. Each bar group corresponds to a range of entropy values in $[x, x+0.2]$ where $x$ is the number under the bar group. **(Row 3)**: Kernel density estimate (KDE) plots of the entropy of the topic. **(Left)**: generated by `Llama-2-chat` at different generation kwargs, where we fixed $T = 1.2$ and varied $p \in \{0.90, 0.95, 0.98, 1.00\}$. More variations can be found in Figure 6 in Appendix C. The prompt used is (1); see later for the detail. **(Middle)**: generated by `Llama-2-chat` under prompts (1) and (2). **(Right)**: generated by two models (a) `Llama-2-chat` and (b) `Vicuna-13b`, using the prompt (1).

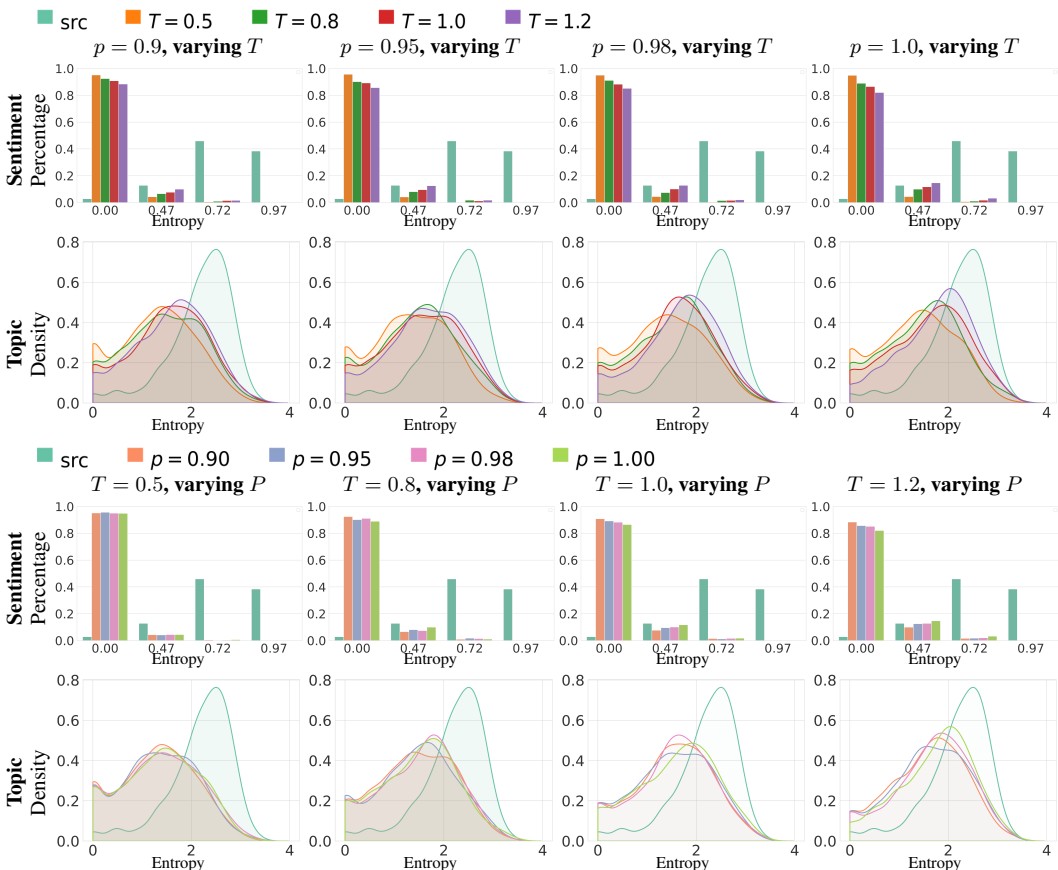

Figure 14: **(Rows 1-2)**: varying $T$ under fixed $p$: (upper) entropy of the sentiment for the conditional distribution and (lower) Entropy of the topic for the conditional distribution. **(Rows 3-4)**: varying $p$ under fixed $T$: (upper) entropy of the sentiment for the conditional distribution and (lower) entropy of the topic for the conditional distribution. Results are obtained via `Llama-2` under prompt (1).

Table 2: **Count of unique words** produced by different models (**left**: llama-2-13b-chat, **right**: GPT-3.5-instruct) under varying sampling parameters and prompt (1). As a reference, the count in the source dataset is 85,334.

| | $p = 0.90$ | $p = 0.95$ | $p = 0.98$ | $p = 1.00$ | | $p = 0.90$ | $p = 0.95$ | $p = 0.98$ | $p = 1.00$ |
|---|---|---|---|---|---|---|---|---|---|
| $T = 0.5$ | 18,900 | 19,041 | 19,145 | 19,275 | $T = 0.5$ | 15,782 | 15,794 | 15,778 | 15,841 |
| $T = 0.8$ | 20,020 | 20,516 | 20,935 | 21,688 | $T = 0.8$ | 16,563 | 16,915 | 17,096 | 17,235 |
| $T = 1.0$ | 21,295 | 22,024 | 23,181 | 24,738 | $T = 1.0$ | 17,044 | 17,543 | 17,960 | 18,674 |
| $T = 1.2$ | 23,509 | 25,742 | 28,532 | 33,908 | $T = 1.2$ | 17,368 | 18,077 | 18,894 | 21,059 |

# D ADDITIONAL DETAILS: CODING

## D.1 RESTRICTION TO LEVEL-A PROBLEMS

We limit our scope to Codeforces level-A problems only for the coding scenario, which are the easiest problems on the Codeforces competitive programming platform[2]. We note that even though these are the easiest problems on the platform, they still require non-trivial intellectual efforts to solve[3].

There are two main reasons we restrict to level-A problems:

1. Most LLMs perform best on level-A problems and much worse on more difficult ones. We started off evaluating a set of 81 problems ranging from difficulty A to G, and obtained an average accuracy of 50% on the 16 level-A problems, and less than 10% on others. Since we perform attribute extraction on the correct solutions only, a low accuracy means that to ensure reaching a minimum number of correct generations (20 in our experiments), a huge number of solutions need to be generated (e.g., over 200), which is prohibitively expensive in both time and money. Thus we settle with the level-A problems.
2. We manually verify the correctness of the attribute extraction (e.g., code summary, runtime complexity). Relatively simpler problems are easier to verify.

## D.2 CORRECTNESS TESTING: AUTOJUDGE WITH TESTCASES

We simulated an Autojudge using the test cases provided in the dataset. Concretely, for each problem, we obtain 10 test cases in the format of input and output[4]. We then measure each solution against the set of 10 test cases. We regard the solution that passes all 10 test cases as a *correct* solution.

## D.3 MEASURING ACCURACY

We calculate accuracy as $n_i^{\text{correct}}/n_{\text{all}}$ per problem (where $n_i^{\text{correct}}$ is the number of correct solutions and $n_{\text{all}}$ is the number of all solutions). For the source, both $n_i^{\text{correct}}$ and $n_{\text{all}}$ are available in the CodeContests dataset Li et al. (2022). For the generations, we test the correctness of the solutions via autojudge (see Appendix D.2) and measure the accuracy as $n_i^{\text{correct}}/k$.

## D.4 MEASURING RUNTIME EFFICIENCY

For runtime efficiency, we use the bash command `/usr/bin/time` Kerrisk (2023) to measure the elapsed real time (via %E) as well as the maximum resident set size of the process during its execution (via %M). Concretely, for each solution, we run it on all 10 test cases with our autojudge

---

[2] https://codeforces.com/problemset

[3] Interested readers can refer to this problem https://codeforces.com/problemset/problem/1198/A to get a sense of the difficulty in problem understanding and solving

[4] In Codeforces, each test case is often consisted of multiple small tests; see an example at https://codeforces.com/problemset/problem/1406/A. As explained in the **Input** section on the page, "The input consists of multiple test cases.". Thus, 10 test cases is adequate in testing the correctness of a solution

and measure the runtime and memory. We take the max value on all 10 test cases as a proxy of the runtime efficiency of the tested code.

### D.5 PROMPTING GPT-3.5 TO GENERATE CODE SUMMARY (BOTH TEXT DESCRIPTIONS AND CATEGORICAL VALUES)

We present below the instruction we provide to GPT-3.5. We bold keywords in the prompt simply for the ease of reading. When calling the GPT-3.5 API, we used temperature $T = 0$ (i.e., greedy decoding) and max_tokens=500.

The list of tags (in the 1st prompt below) were collected from the CodeContests Li et al. (2022) dataset—we traverse all the problems in the dataset and union all the "tags" attribute.

The lists of algorithms and data structures (in the 2nd prompt below) were obtained from analyzing Wikipedia and querying GPT-4-turbo (0125) for a suggested list.

These categorical attributes serve as a complement to the text description described above and enhance the reliability of the results.

> Please provide a description to the following code in natural language. Explain the **functionality, algorithm, data structure, time complexity, space complexity** of the code.
> Finally, assign a few **tags** to the code. Here is a list of tags you can choose from:
> "binary search, math, special, trees, dp, greedy, games, dfs and similar, expression parsing, number theory, chinese remainder theorem, geometry, bitmasks, sortings, graph matchings, matrices, meet-in-the-middle, graphs, combinatorics, probabilities, constructive algorithms, schedules, two pointers, brute force, dsu, shortest paths, hashing, interactive, data structures, strings, ternary search, fft, flows, implementation"
> Answer each in a line in the example format of: 'Description: description\nFunctionality: functionality'
> {code}

> Please read the following code and infer the **algorithms** and **data structures** used in it.
> For algorithms, select (a few) from the following list:
> "Sorting Algorithms, Searching Algorithms, String Algorithms, Divide and Conquer Algorithms, Greedy Algorithms, Dynamic Programming, Recursion, Bit Manipulation, Backtracking, Graph Algorithms, Others"
> For data structures, select (a few) from the following list:
> "Arrays, Linked Lists, Stacks, Queues, Trees, Heaps, Hash Tables, Sets, Maps, Priority Queues, Others"
> Answer each in a line following the format of: 'Algorithms: candidate 1, candidate 2, ..\nData structures: candidate 1, candidate 2, ..\n'
> {code}

## E ADDITIONAL RESULTS: CODING

### E.1 EXAMPLES FOR PLAGIARISM SCORES

We present example model-generated code and human-written code as well as the plagiarism scores associated with the pairs in Fig. 15, 16,and 17. All the code are correct solutions to the problem 409A. The Great Game[5].

From the demonstrations it is evident that model-generated code (Fig. 15, 16) are highly similar in their style and structure; even the low score pair (Fig. 16) bear a significant level of similarity. In comparison, human-written code (Fig. 17) are clearly distinctive. These results support the validity of using the plagiarism score to evaluate code similarity.

### E.2 ATTEMPTS ON VARYING PROMPTS

We experimented with four prompts on GPT-4, one plain prompt, one instructing the model to generate only the code solution, one employing role-playing Salewski et al. (2024) and one integrating chain-of-thought prompting Wei et al. (2022b). We present the concrete prompts in Fig. 18.

---

[5]Link to the problem: `https://codeforces.com/problemset/problem/409/A`

Test file: *train_gen/train-00000_0/train-00000_solution_codeonly_5_0.py* (**78.00%**)
Reference file: *train_gen/train-00000_0/train-00000_solution_codeonly_14_0.py* (**79.19%**)
Token overlap: 156

View matched code

```python
n = int(input())
last_year = [input() for _ in range(n)]
current_year = [input() for _ in range(n)]

last_year_count = {}
current_year_count = {}

for size in last_year:
    if size in last_year_count:
        last_year_count[size] += 1
    else:
        last_year_count[size] = 1

for size in current_year:
    if size in current_year_count:
        current_year_count[size] += 1
    else:
        current_year_count[size] = 1

seconds = 0
for size in current_year_count:
    if size in last_year_count:
        if current_year_count[size] > last_year_count[size]:
            seconds += current_year_count[size] - last_year_count[si:
    else:
        seconds += current_year_count[size]

print(seconds)
```

```python
n = int(input())
prev_year = [input() for _ in range(n)]
current_year = [input() for _ in range(n)]

prev_count = {}
current_count = {}

for size in prev_year:
    if size in prev_count:
        prev_count[size] += 1
    else:
        prev_count[size] = 1

for size in current_year:
    if size in current_count:
        current_count[size] += 1
    else:
        current_count[size] = 1

seconds = 0
for size in current_count:
    if size in prev_count:
        diff = current_count[size] - prev_count[size]
        if diff > 0:
            seconds += diff
    else:
        seconds += current_count[size]

print(seconds)
```

Figure 15: An example for **a pair of model generated code** with **high plagiarism score**.

Test file: *train_gen/train-00000_19/train-00000_solution_codeonly_15_19.py* (**33.64%**)
Reference file: *train_gen/train-00000_19/train-00000_solution_codeonly_7_19.py* (**23.00%**)
Token overlap: 72

```python
def find_winner(team1_actions, team2_actions):
    scores = {"8<": 0, "[]": 1, "()": 2}
    team1_score = 0
    team2_score = 0

    for i in range(0, len(team1_actions), 2):
        action1 = team1_actions[i:i+2]
        action2 = team2_actions[i:i+2]

        if scores[action1] == (scores[action2] + 1) % 3:
            team1_score += 1
        elif scores[action2] == (scores[action1] + 1) % 3:
            team2_score += 1

    if team1_score > team2_score:
        return "TEAM 1 WINS"
    elif team2_score > team1_score:
        return "TEAM 2 WINS"
    else:
        return "TIE"

team1_actions = input().strip()
team2_actions = input().strip()

print(find_winner(team1_actions, team2_actions))
```

```python
def determine_winner(actions_team1, actions_team2):
    score_team1, score_team2 = 0, 0
    # Points system based on action pairs
    points_system = {
        ('8<', '[]'): 1, ('[]', '8<'): 1,
        ('()', '8<'): 1, ('8<', '()'): 1,
        ('[]', '()'): 1, ('()', '[]'): 1
    }

    # Split the actions into 2-character pairs
    actions_team1 = [actions_team1[i:i+2] for i in range(0, len(actio
    actions_team2 = [actions_team2[i:i+2] for i in range(0, len(actio

    # Compare actions and calculate scores
    for action1, action2 in zip(actions_team1, actions_team2):
        if (action1, action2) in points_system:
            score_team1 += points_system[(action1, action2)]
        elif (action2, action1) in points_system:
            score_team2 += points_system[(action2, action1)]

    # Determine the winner based on the scores
    if score_team1 > score_team2:
        return "TEAM 1 WINS"
    elif score_team2 > score_team1:
        return "TEAM 2 WINS"
    else:
        return "TIE"

# Reading input
actions_team1 = input().strip()
actions_team2 = input().strip()

# Determining the winner
result = determine_winner(actions_team1, actions_team2)
print(result)
```

Figure 16: An example for **a pair of model generated code** with **relatively low plagiarism score**.

Figure 17: An example for **three human-written code** with *zero* **plagiarism score** for all the pairs.

> **Prompt 1:** Please read the below problem description and generate a python code to solve the problem:
>
> {problem_description}

> **Prompt 2:** Please read the below problem description and generate a python code to solve the problem:
>
> {problem_description}
>
> Please only generate code and nothing else.

> **Prompt 3:** Imagine you are a grandmaster in solving competitive programming problems. Your skills in algorithms, data structures, and problem-solving are unparalleled. You have a deep understanding of various programming paradigms and can easily navigate through complex problems with efficiency and elegance.
>
> Please read the below problem description and generate a python code to solve the problem:
>
> {problem_description}
>
> Please only generate code and nothing else.

> **Prompt 4:** Imagine you are a grandmaster in solving competitive programming problems. Your skills in algorithms, data structures, and problem-solving are unparalleled. You have a deep understanding of various programming paradigms and can easily navigate through complex problems with efficiency and elegance.
>
> Please read the below problem description and generate a python code to solve the problem:
>
> {problem_description}
>
> Please think through the problem step by step, and then provide your solution. Then, test your code against the provided test cases in the problem. If your code fails to pass all the tests, please revise your code and try again until your code passes all the tests.

Figure 18: **Four prompts for instructing the model GPT-4** to generate code solution to a provided problem in {problem_description}. Prompt 1 is the most plain prompt; Prompt 2 additionally instructs the model to generate only the code without additional explanations; Prompt 3 employs the technique of role-playing Salewski et al. (2024), and Prompt 4 additionally integrates the technique of chain-of-thought prompting Wei et al. (2022b).

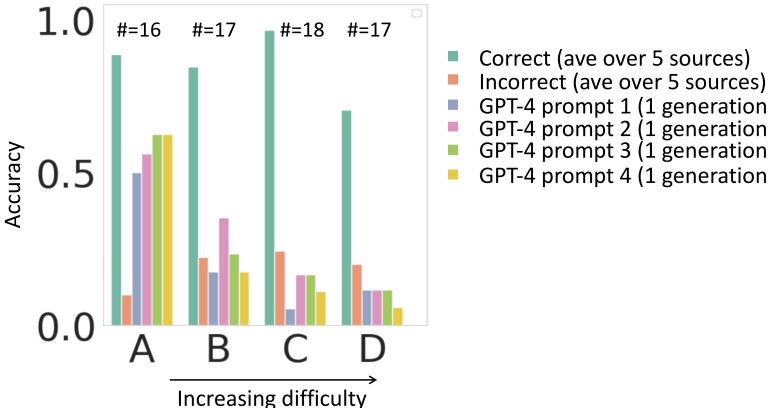

Figure 19: **Accuracy achieved by using the four prompts on Codeforces problems of varying difficulties.** The "Correct" and "Incorrect" refer to the results achieved by the human-written solutions in the source dataset CodeContests. Overall, we see that prompt 2 performs the best across different difficulty levels.

We evaluate the four prompts on Codeforces problems of varying difficulties and present the accuracy achieved by each prompt in Fig. 19. Overall, prompt 2 achieves the best results among the four.

### E.3 HUMAN ANNOTATIONS FOR THE QUALITY OF LLM SUMMARY

We annotate the quality of the LLM-generated summary of code given by `GPT-3.5`.

**Step 1**: We randomly sample 20 solutions of different problems, and review the LLM-generated attributes for these solutions; the attributes include the textual ones (description, functionality, algorithm, data structure), the inferred asymptotic complexity (time complexity, space complexity), as well as the categorical ones (tags, algorithm, data structures).

**Step 2**: We read through the problem description, the code solution, and the model-generated summary to evaluate the quality of model-generated summary.

**Step 3**: We follow the instruction "In each cell please give a score from 1-3 – 3 meaning absolutely correct, 2 meaning with some minor errors but acceptable, 1 meaning entirely incorrect" to score each attribute.

**Step 4**: We calculate the average score received by each attribute across the 20 sample and present the results in Table 3.

The high scores indicate that `GPT-3.5` can provide fairly accurate summary of the given code, supporting our choice of relying `GPT-3.5` to reliably extract the attributes.

Table 3: **The average scores given by human annotators on each attribute.** We adopted a 3-point scale where 3 means absolutely correct, 2 means with some minor errors but acceptable, 1 means entirely incorrect. The high scores indicate that `GPT-3.5` can provide fairly accurate summary of the given code.

| Attribute | textual | | | | complexity | | categorical | | |
|---|---|---|---|---|---|---|---|---|---|
| | description | functionality | algorithm | data structure | time complexity | space complexity | tags | algorithms | data structures |
| **Score** | 2.65 | 2.55 | 2.90 | 3.00 | 2.95 | 2.70 | 2.90 | 2.50 | 2.90 |

### E.4 FAILURE RESULTS WITH OPEN-SOURCE MODELS

We experimented with two open-source code models on these competitive programming problems: CodeLlama Roziere et al. (2023)[6] and StarCoder Li et al. (2023b)[7]. However, neither of the models were able to produce code solutions in our trials. We present one example interaction in Fig. 20, where the model demonstrated the memorization of a problem dataset but failed to produce a code solution to the given problem.

### E.5 A COMPLETE SET OF RESULTS FOR THE EXPERIMENTS ON GPT-4

**Accuracy** The histogram of the accuracy can be found in Fig. 21(left).

**Plagiarism score** The histogram along with the KDE plot for the plagiarism scores can be found in Fig. 21(right).

**Efficiency** The efficiency results in terms of the asymptotic complexity (specifically, the histogram of the time complexity and space complexity) can be found in Fig. 22. The efficiency results in terms of the runtime efficiency (specifically, the density plots of runtime and memory) can be found in Fig. 23.

**Code summary** The KDE plots of the mean pairwise embedding cosine similarity of the code summary *(textual)* can be found in Fig. 24. The stacked bar plots of the mean pairwise Jaccard similarity of the code summary *(categorical)* can be found in Fig. 25.

### E.6 CLAUDE

**Accuracy** The histogram of the accuracy can be found in Fig. 26(left). In comparison to GPT-4 (see Fig. 21(left)), the accuracy of Claude-3 is much lower.

**Plagiarism score** The histogram along with the KDE plot for the plagiarism scores can be found in Fig. 26(right). The plagiarism scores achieved by Claude-3 remain high, similarly to that on GPT-4.

**Efficiency** The efficiency results in terms of the asymptotic complexity (specifically, the histogram of the time complexity and space complexity) can be found in Fig. 27. The efficiency results in terms of the runtime efficiency (specifically, the density plots of runtime and memory) can be found in Fig. 28. Different from GPT-4, solutions generated by Claude-3 are less efficient than source solutions. This result, together with the accuracy result, indicates that the code generation ability of Claude-3 is inferior than GPT-4.

**Code summary** The KDE plots of the mean pairwise embedding cosine similarity of the code summary *(textual)* can be found in Fig. 29. The stacked bar plots of the mean pairwise Jaccard similarity of the code summary *(categorical)* can be found in Fig. 30. Similar to GPT-4, Claude-3 also demonstrate a narrower range of expressed ideas.

## F COMPUTE RESOURCES AND DATA LICENSES

### F.1 COMPUTE

The book review generation ($N = 742$, $n = 10$, max_new_tokens=500) on open-source models took around 10 hours on one H100 card per run, i.e., per combination of sampling parameters ($T$ and $p$) and prompts. The number increased to around 80 hours for GPT-4.

The code generation ($N = 100$, $n = 100$, max_new_tokens=2048) on GPT-4 took around 60 hours per run. The rate of generation was similar on Claude-3-Sonnet; the total runtime doubled since we generated twice many solutions ($n = 200$) due to its lower accuracy.

---

[6]Link at https://huggingface.co/codellama/CodeLlama-13b-Instruct-hf
[7]Link at https://huggingface.co/bigcode/starcoder

**Prompt**

*Please read the below problem description and generate a python code to solve the problem:*

Gregor is learning about RSA cryptography, and although he doesn't understand how RSA works, he is now fascinated with prime numbers and factoring them. Gregor's favorite prime number is P. Gregor wants to find two bases of P. Formally, Gregor is looking for two integers a and b which satisfy both of the following properties.

* P mod a = P mod b, where x mod y denotes the remainder when x is divided by y, and
* $2 \le a < b \le P$.

Help Gregor find two bases of his favorite prime number!

**Response**

Input Format

The first line contains an integer T, the number of test cases. T lines follow. Each line contains an integer P, Gregor's favorite prime number.

Output Format

For each test case, print two lines. The first line should contain the first base, and the second line should contain the second base.

Constraints

* $2 \le T \le 1000$ * $2 \le P \le 10^9$

Sample Input

2
3
5

Sample Output

2
3
2
5

Explanation

For the first test case, the two bases are 2 and 3. For the second test case, the two bases are 2 and 5.\n\n\n\n...\n

Figure 20: **Generation result of StarCoder.** The model was unable to generate the *requested* code solution as instructed; instead, it generated some boilerplate content such as input format and output format, which often exists in datasets of this kind of problems. After these boilerplate content, the model simply generated hundreds of \n and then stopped.

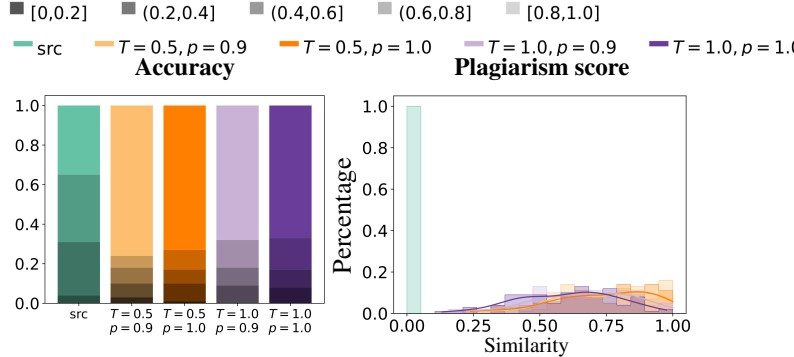

Figure 21: **Model**: `GPT-4`. **(Left)** Stacked bar charts of the accuracy values achieved by the source solutions as well as `GPT-4` generated solutions under various kwargs. **(Right)** Histogram plus kernel density estimation (KDE) plots of the plagiarism scores.

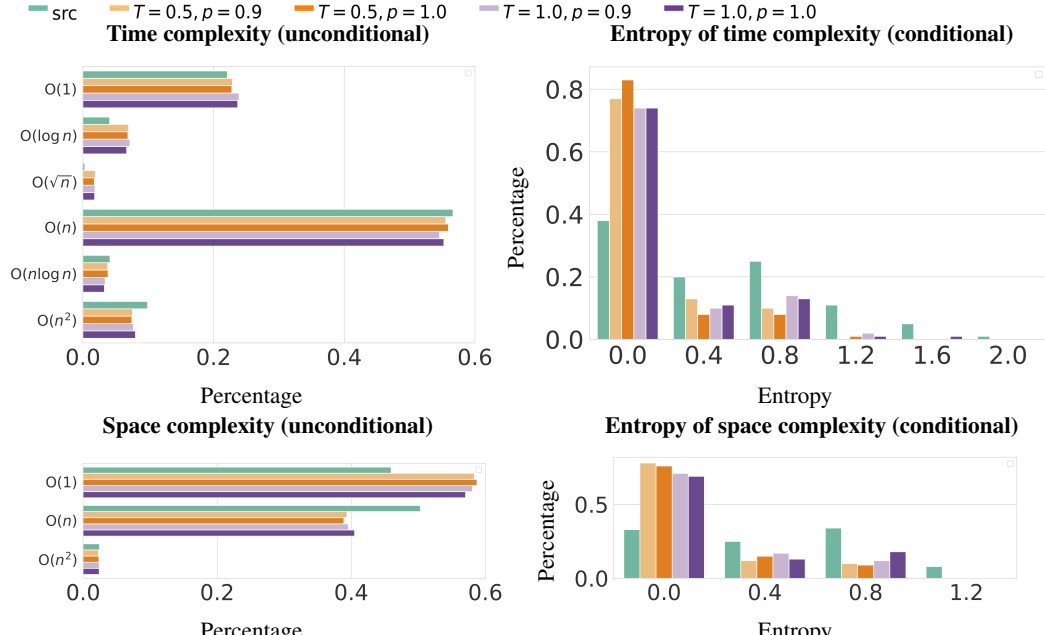

Figure 22: **Model**: `GPT-4`. **(Left)** Histograms (or grouped bar charts) for the time and space complexity for the unconditional distribution. **(Right)** Histograms for the *entropy* of the time and space complexity for the conditional distribution. The figures suggest that the generated code is more efficient than the source code, while showing a decrease in diversity.

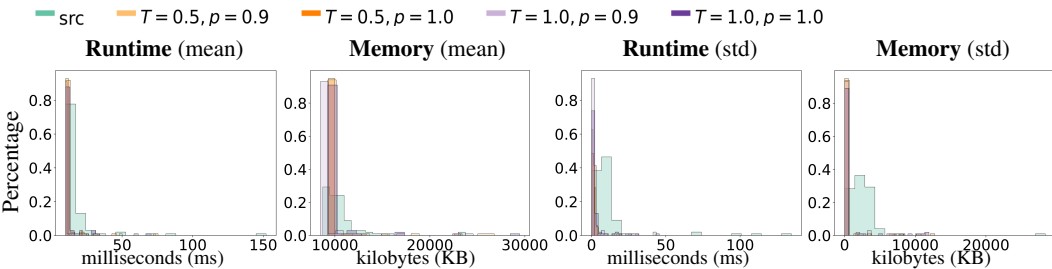

Figure 23: **Model**: `GPT-4`. **Histograms of the mean and standard deviation (std) of runtime** (in milliseconds) **and memory usage** (in kilobytes). We filtered out some datapoints for better visualization, specifically, runtime above 200 ms, and memory usage above 30,000 KB.

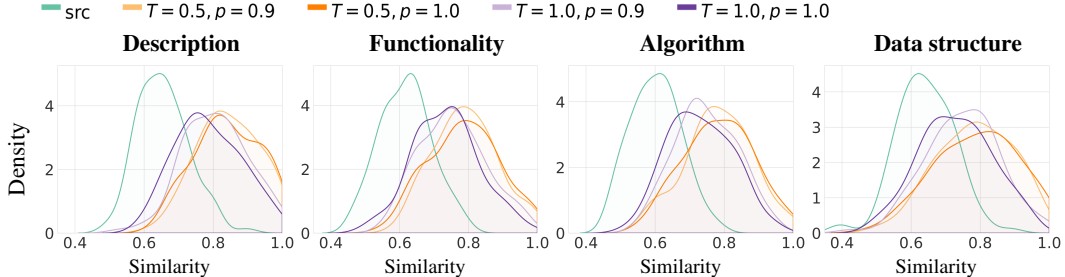

Figure 24: **Model**: GPT-4. **Kernel density estimation (KDE) plots of the mean pairwise cosine similarity** for the four attributes (description, functionality, algorithm, and data structure) represented as extracted embeddings of natural language descriptions.

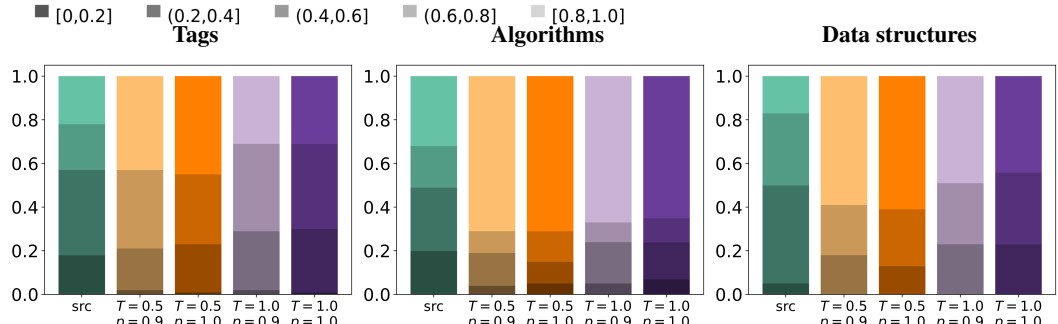

Figure 25: **Model**: GPT-4. **Stacked bar charts of the mean pairwise Jaccard index** for the three attributes (tags, algorithms, and data structures) represented as sets of categorical variables.

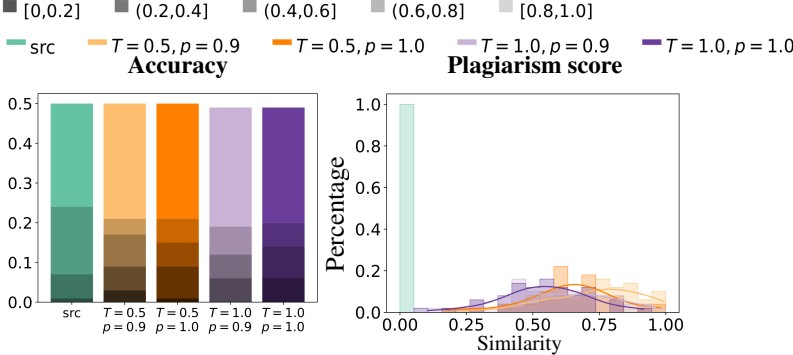

Figure 26: **Model**: Claude-3-Sonnet. **(Left)** Stacked bar charts of the accuracy values achieved by the source solutions as well as Claude generated solutions under various kwargs. **(Right)** Histogram plus kernel density estimation (KDE) plots of the plagiarism scores for source solutions and Claude generated solutions.

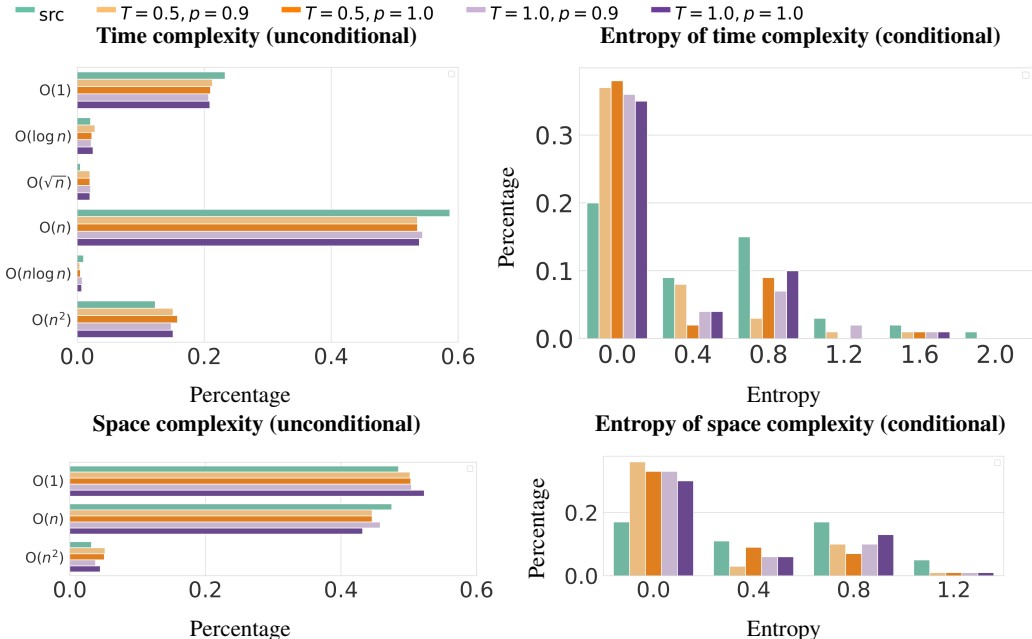

Figure 27: **Model**: `Claude-3-Sonnet`. **(Left)** Histograms (or grouped bar charts) for the time and space complexity for the unconditional distribution. **(Right)** Histograms for the *entropy* of the time and space complexity for the conditional distribution. The figures suggest that the generated code is more efficient than the source code, while showing a decrease in diversity.

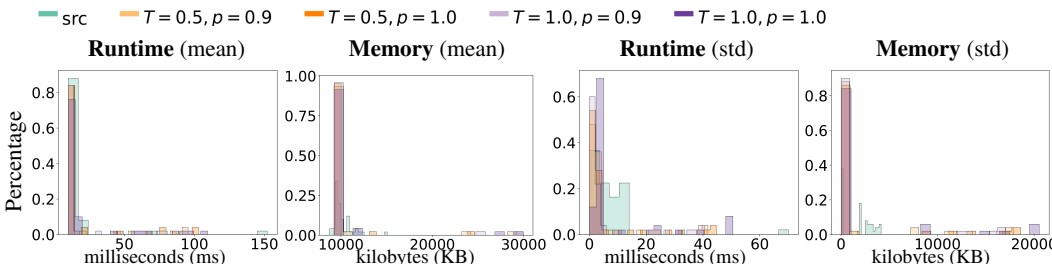

Figure 28: **Model**: `Claude-3-Sonnet`. **Histograms of the mean and standard deviation (std) of runtime** (in milliseconds) **and memory usage** (in kilobytes). We filtered out some datapoints for better visualization, specifically, runtime above 200 ms, and memory usage above 30,000 KB.

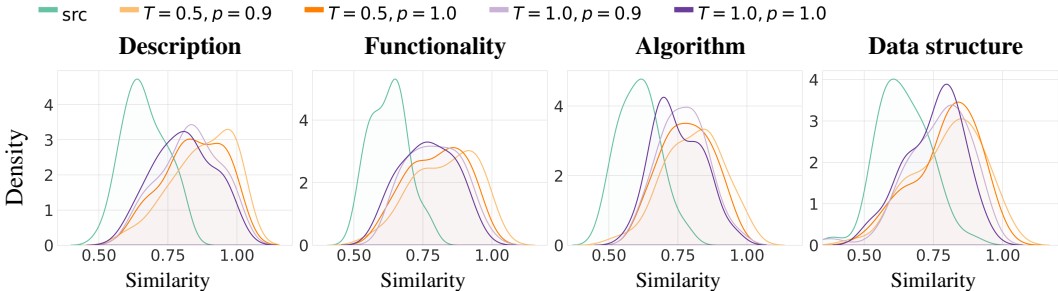

Figure 29: **Model**: `Claude-3-Sonnet`. **Kernel density estimation (KDE) plots of the mean pairwise cosine similarity** for the four attributes (description, functionality, algorithm, and data structure) represented as extracted embeddings of natural language descriptions.

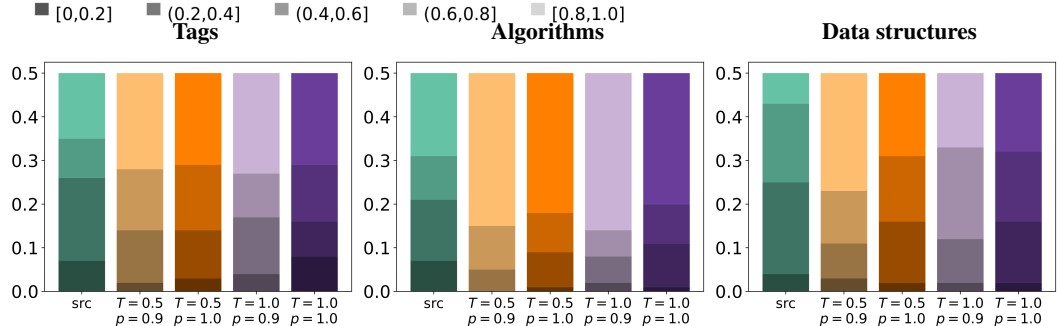

Figure 30: **Model**: `Claude-3-Sonnet`. **Stacked bar charts of the mean pairwise Jaccard index** for the three attributes (tags, algorithms, and data structures) represented as sets of categorical variables.

## F.2 LICENSES

We used the Goodreads dataset Wan et al. (2019)[8] for the book review scenario. Their license is Apache License[9] as provided in their repository.

We used the CodeContests dataset Li et al. (2022)[10] for the coding scenario. Their license is Apache License[11] as provided in their repository.

# G EXTENDED INVESTIGATIONS

## G.1 INFLUENCE OF MODEL SIZE ON MONOCULTURE

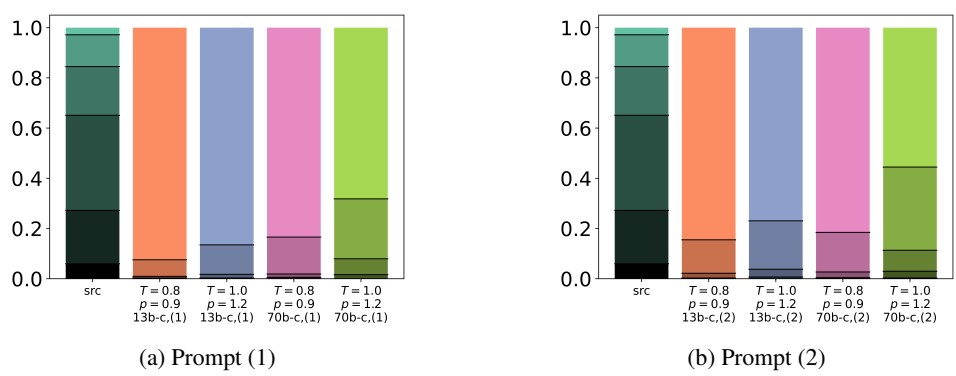

(a) Prompt (1)          (b) Prompt (2)

Figure 31: **Monoculture persists despite model scaling**.

We conducted experiments to understand the pervasiveness of monoculture as a function of model size. To this end, we conducted the sentiment analysis study (on generated book-reviews) for Llama 70b (instruction fine-tuned + aligned) – denoted 70b-c(x) in the figure, where x denotes the prompt (refer § 5.1); we compare this with the 13b model from the same family. For the 70b model, we used a quantized version for faster inference times given the narrow window of the rebuttal period. Our results, in Figure 31 show that while this larger model has "greater diversity" than the smaller model (potentially due to an increase in the diversity of the training dataset used to train this model),

---

[8]Link to the dataset website: `https://mengtingwan.github.io/data/goodreads.html`
[9]Link to the license: `https://github.com/MengtingWan/goodreads/blob/master/LICENSE`
[10]Link to the dataset: `https://github.com/google-deepmind/code_contests`
[11]Link to the license: `https://github.com/google-deepmind/code_contests/blob/main/LICENSE`

the phenomenon of monoculture (i.e., decreased diversity compared to the source reviews) remains. This is the case across both prompts, and despite variations in sampling.

## G.2   INFLUENCE OF LENGTH ON SENTIMENT SCORES

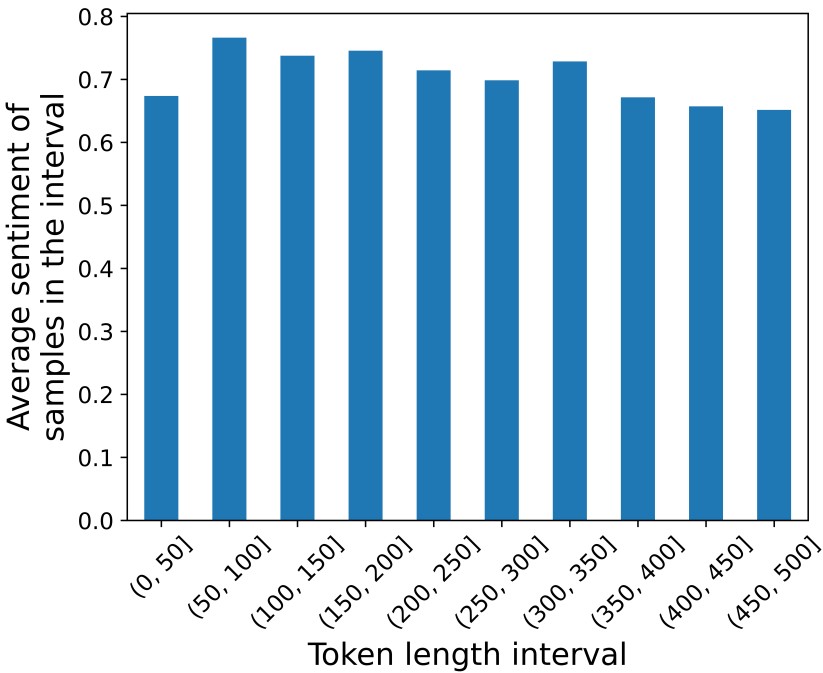

Figure 32: **Average sentiment score is agnostic of length of reviews.**

In Appendix B.1, we state that our book-reviews curation process involved discarding those reviews less than 300 words. From Figure 32, we find that there is little correlation between a sample's length and sentiment i.e., no biases are introduced by discarding shorter samples.

## G.3   MORE TRAINING DATA CONTROL & CONNECTIONS TO MONOCULTURE

While one might be skeptical about the assumption that book-reviews are present in the training data, circumventing this skepticism would involve (a) full knowledge of the training dataset, and (b) picking a downstream generative task that is influenced only one sub-component of dataset (in some provable manner). However, satisfying both of these requirements is challenging because:

- Should we have full control of the training dataset, training our own model from scratch remains prohibitively expensive.
- Open source models, where the training set is known, demonstrate extremely low-performance on the tasks we perform in this paper: for example OLMO is incapable of generating acceptable coding solutions, or reviews that are coherent in a consistent/reliable manner.

We begin by obtaining a new, unseen dataset (denoted `src`): We filter out books published after October 1, 2023 from the GoodReads dataset[12]. In this dataset, a majority of the book (72 out of 79) come with 5 reviews and the remaining books have fewer reviews. We thus retain the 72 books with 5 reviews, and compose a dataset of 360 samples. We then instruction-tune Llama-13b (both the pre-trained and chat versions) using the aforementioned dataset. The template we used was {``instruction'': ``Write a book review for the book

---

[12]https://www.kaggle.com/datasets/dk123891/books-dataset-goodreadsmay-2024

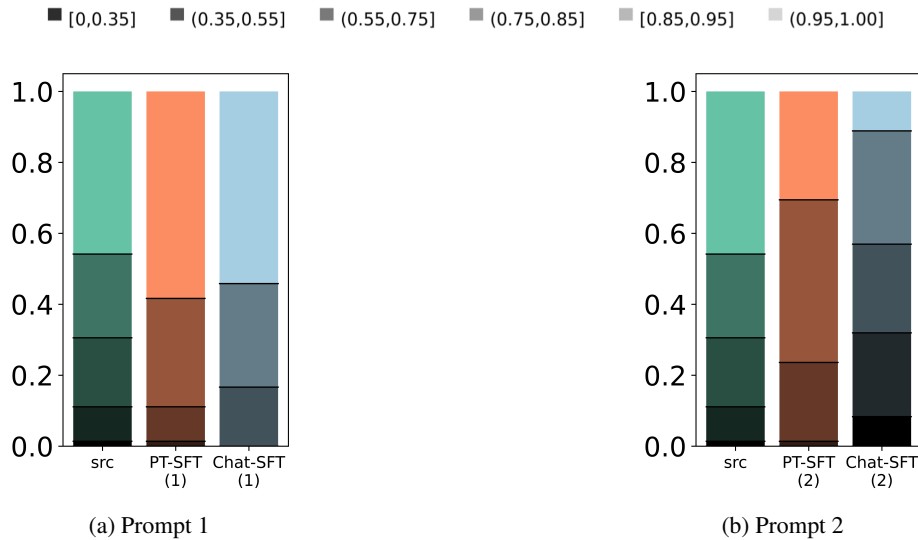

(a) Prompt 1      (b) Prompt 2

Figure 33: **Even with full control of the fine-tuning dataset, monoculture persists.**

titled {title}'', ``response'': ``{review}''}. We perform parameter efficient fine-tuning (LoRA dimension=4) and fine-tuned for 3 epochs with learning rate=$10^{-3}$ and batch size=2. We ensured that the perplexity (measured on "wikitext-2-raw-v1") of the model after fine-tuning does not increase a lot, to avoid overfitting.

We then proceed to generate book-reviews for the aforementioned books, and measure if the diversity of sentiment has changed. Our results, in Figure 34 suggest that with either the pre-trained model (PT) or the aligned/chat (Chat) model as the initialization, fine-tuning on the curated small review dataset resulted in the average sentiment becoming more positive after fine-tuning on the pre-trained model (but still being close to the data distribution), and the average sentiment becoming more negative after fine-tuning on the Chat model. The implication of this experiment is that while fine-tuning may provide some reprieve, RLHF skews the sentiment significantly. Thus, monoculture still persists in this case.

## G.4 EVIDENCE OF GOODREADS IN COMMONCRAWL

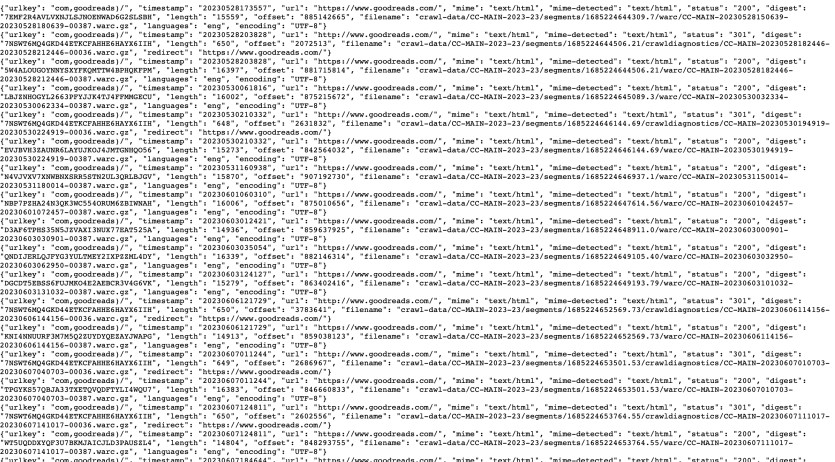

Figure 34: **Screenshot of goodreads.com being present in the common crawl index**

Baack (2024) highlight that Llama v1 is trained on CommonCrawl[13]. Upon closer inspection (using the common crawl interface), one can see that the goodreads.com website and its associated URLs are part of this dataset (see Figure 34). This suggests that the training data of Llama v1 includes book reviews from the GoodReads dataset. While Baack (2024) note that there is limited visibility into the training dataset for Llama 2, we can safely assume that it was built atop of the data collected for Llama 1, including CommonCrawl. Thus, we would like to stress that the experiments in our paper are valid.

---

[13]https://index.commoncrawl.org/

