# OpenReview forum: "Generative Monoculture in Large Language Models"
_ICLR.cc/2025/Conference — ICLR 2025 Poster_

### Official Review · Reviewer_L9QX · 2024-10-29

**Soundness:** 2
**Presentation:** 4
**Contribution:** 2
**Rating:** 5
**Confidence:** 4

**Summary:**

The paper investigates generative monoculture in LLMs, defined as the reduction in diversity from training data to the model’s outputs. It examines this phenomenon in two tasks: sentiment of book reviews and code generation. The paper found consistent monoculture in all experiments.

**Strengths:**

Generative monoculture is a subtle phenomenon that may not be immediately noticeable to users but can have a lasting impact on society, making it an important area of study. The paper is well-written and clearly presented.

**Weaknesses:**

W1: One major risk in this study is that measuring generative monoculture with “a source dataset that is likely to have been used in training the LLM we wish to investigate” (line 128) could introduce substantial inaccuracies. This approach risks significant biases, undermining the validity of the measurements. I will elaborate below and hope the authors could discuss these risks and present evidence to support validity of their approach.

W1-1: First, the selected source data may not actually be part of the LLM’s training data.

W1-2: Even if the selected source dataset is indeed in the training data, it may not represent the full range of relevant training data. For instance, other sources or non-English reviews of the same books could dominate the distribution from training data, meaning that the measured distribution may not reflect the true distribution of the training data at all. This concern is especially relevant since the study uses relatively small datasets (742 books with 10 reviews per book for book reviews and 100 coding questions with 20 correct solutions each for code).

W1-3: Additionally, the filtering of the source dataset introduces biases. The authors mention in Appendix B2 that book reviews are filtered to be between 300-700 words. Filtering by length could affect the sentiment distribution, potentially skewing the results.

W2: Could the authors clarify why they filter out low-quality (generated) book reviews by examining perplexity? While low-quality reviews may seem less helpful to users, excluding them might introduce bias in the sentiment distribution of LLM-generated book reviews. I recommend testing whether this filtering step influences the measured sentiment distribution, although it might be preferable to retain these low-quality reviews to avoid introducing additional bias.

W3: To strengthen the study’s validity, I suggest measuring generative monoculture in a more controlled environment, where the training data is known and the training distribution can be accurately measured. Without certainty that the source dataset is part of the training data, all findings are at risk of being unreliable.

W4: In Line 317, the efficiency of LLM generated code is measured by ``prompt GPT-3.5 to infer the big O time and space complexity’’. Please establish the reliability of this LLM evaluation.

W5: In line 328, Code Summary (categorical) is measured by “prompt GPT-3.5 to assign tags to a code segment by providing it a set of tags to choose from”. Please establish the reliability of this LLM evaluation.

W6: In Figure 4 (d), decay T p = 1.0 seems to have higher entropy, therefore more diverse, than source distribution. Could the authors discuss this, as it disagrees with the statement in line 379 “there exists significant narrowing from the source to generation distribution in all attributes considered for both scenarios”?

**Questions:**

Q1: Could the authors explain how the plagiarism score is computed in Figure 5 (b)? It seems is not mentioned in paper.

---

> ### Author Response · Authors · 2024-11-20
> **Thank you for your feedback! (part 1)**
>
> Thank you for your thoughtful comments on our paper.
>
> **(W-1-1):** While it is unlikely that models we consider do not use the datasets we consider given the extreme breadth of data scraped from the internet to train most LLMs, we can not provide definitive proof of their presence given the closed nature of these models, and the lack of reliable dataset detection methods [1].
>
> **(W-1-2):**  We appreciate this point. We believe the goodreads dataset is our attempt at an approximation of a sample of the book review training data, and while there likely is some deviation from the true distribution of training data (which is unknown and impossible to determine for these models), we think there is good reason to believe our main point still stands: note just how skewed the LLM-generated book reviews are towards positive sentiment: both GPT3.5 and 4 had returned 100% of their responses rated in the most positive bucket under prompt (1), and never returned less than 97.6% in the most positive bucket (see Takeaway 1). It is hard to fathom that the overall (unknown) distribution of book reviews is this skewed, with next to no negative samples— especially given that the goodreads data is a part of the training data as confirmed above and has data in this range. This suggests that there is at least some part of the distribution that has strong negative sentiment, and since we have evidence of no strong negative sentiment generations, we believe this suggests that generative monoculture is occurring.
>
> **Impact of source filtering (W1-3)** Upon review, we find that there is little correlation between a sample’s length and sentiment i.e., no biases are introduced. Please find the figure in Appendix H of the revised draft.
>
> **Impact of target filtering (W2)** We filter out low-quality generated results because they were illegible (gibberish), and could not be recognized as a book review. Similar to only wanting to compare diversity on correct code solutions, we only want to compare sentiment on generations that are recognizable as a book review. We adopted the reviewer’s suggestion to analyze whether this filtering step would introduce additional bias; it turned out that the change in average sentiment score is on the magnitude of 0.001, meaning that the impact of filtering on the sentiment is negligible.
>
> We have also provided some examples of book reviews at different perplexity scores in Appendix C.1.  Since we could not manually inspect all outputs, we investigated random samples of generations at various perplexity scores and decided on a threshold that admitted the most generations possible while still being comprehensible. Another reason we looked into perplexity is to understand the practicality of high randomness generation as a mitigation—as we presented in Table 1 and explained at line 463; although high randomness could lead to improved diversity, the generation quality degraded to a point where it could not be adopted for practical use.

---

> > ### Author Response · Authors · 2024-11-20
> > **(part 2)**
> >
> > **Measuring generative monoculture in a controlled env (W3)** We agree with the review that this would be the ideal experiment, and after much deliberation decided to use the experimental design in our paper to best navigate the following obstacles discussed below:
> >
> > 1. Training our own model from scratch is prohibitively expensive.
> > 2. Open source models, where the training set is known, demonstrate extremely low-performance on the tasks we perform in this paper: for example OLMO is incapable of generating acceptable coding solutions, or reviews that are coherent in a consistent/reliable manner.
> >
> > We begin by obtaining a new, unseen dataset: We filter out books published after October 1, 2023 from the GoodReads dataset (May 2024 ver). In this dataset, a majority of the book (72 out of 79) come with 5 reviews and the remaining books have fewer reviews. We thus retain the 72 books with 5 reviews, and compose a dataset of 360 samples. We then instruction-tune Llama-13b (both the pre-trained and chat versions) using the aforementioned dataset. The template we used was {“instruction”: "Write a book review for the book titled {title}", “response”: “{review}”}. We perform parameter efficient fine-tuning (lora dimension=4) and fine-tuned for 3 epochs with learning rate=1e-3 and batch size=2. We ensured that the perplexity (measured on “wikitext-2-raw-v1”) of the model after fine-tuning does not increase a lot, to avoid overfitting.
> >
> > We then proceed to generate book-reviews for the aforementioned books, and measure if the diversity of sentiment has changed. Our results suggest that with either the pre-trained model or the aligned/chat model as the initialization, fine-tuning on the curated small review dataset resulted in the average sentiment becoming more positive after fine-tuning on the pre-trained model (but still being close to the data distribution), and the average sentiment becoming more negative after fine-tuning on the Chat model. The implication of this experiment is that while fine-tuning may provide some reprieve, RLHF skews the sentiment significantly.
> > These results are presented in Appendix I of the revised draft.
> >
> >
> >
> >
> > **Reliability of LLM evaluation (W4+W5)** To alleviate the concern of lack of reliability of evaluation by GPT 3.5, the authors on this paper have (independently) manually validated the quality of the responses. Detailed procedures and results can be found in Appendix E.3 and Table 3, referenced at line 313 in main paper. The scores in Table 3 show that the extracted results given by GPT-3.5 are highly accurate and reliable. In addition to the attributes extracted by GPT-3.5, we also included attributes obtained from other means to serve as a supplement wherever possible, e.g., for measuring “efficiency”, in addition to the “complexity” results provided by GPT-3.5, we also measured the runtime and memory usage as a support. We hope these together increase the credibility of the results.
> >
> > **Topic entropy (W6).** We clarify about the interpretation of Fig. 4 (d). We present a KDE plot of entropy, where more mass on the higher end (right side) means more diverse (the green curve for src), while more mass on the lower end (left side) means less diverse (particularly, the dark orange and purple curve for decay T). If we examine particular points, say, entropy=0 on the figure (meaning no diversity at all), then for the dark orange curve (decay T, p=1.0, (1)), there are more than 20% of the samples where there’s no topic diversity in the response at all, which is astoundingly high.
> >
> > **Calculation of the plagiarism score (Q1).** Plagiarism score is essentially the mean pairwise ﬁngerprint similarity (line 337-338, and Table 3), where the “fingerprint” of a program is a selected set of hash values generated by the winnowing algorithm [2], and the “similarity” between two programs is measured by comparing the overlap of their fingerprints. Implementation-wise, we reply on an existing tool CopyDetect to obtain the similarity score. We provided the description of the calculation in Sec 3.2 line 199-205. Please let us know if anything is unclear.
> >
> > [1] Roberts, Manley, Himanshu Thakur, Christine Herlihy, Colin White, and Samuel Dooley. "To the cutoff... and beyond? a longitudinal perspective on LLM data contamination." In The Twelfth International Conference on Learning Representations. 2023.
> >
> > [2] Schleimer, Saul, Daniel S. Wilkerson, and Alex Aiken. "Winnowing: local algorithms for document fingerprinting." Proceedings of the 2003 ACM SIGMOD international conference on Management of data. 2003.
> >
> > We hope that we have satisfactorily responded to your queries, and you would consider raising your score for this work.

---

> > > ### Author Response · Authors · 2024-11-22
> > > **Happy to engage!**
> > >
> > > Thank you for the wonderful questions; we are happy to engage during the rebuttal period and clarify any concerns! Being the reviewer with the lowest score, your perception and feedback are very important to us!

---

> > > ### Comment · Reviewer_L9QX · 2024-11-22
> > >
> > > Regarding the authors's response to W3 in part 2.
> > >
> > > 1. I was surprised that OLMO struggled to consistently respond to queries like "Write a personalized review for the book titled {title}." But I understand this is out of the authors' control.
> > >
> > > 2. I appreciate the additional experiment but find it challenging to understand the implications. Could the authors help clarify?
> > > a. takeaway 3 says RLHF hurts diversity. so chat model is less diverse than PT models.
> > > b. for prompt 1 and 2 in figure 33, Chat-SFT becomes more diverse than PT-SFT?
> > > c. so "fine-tuning may provide some reprieve", but why would it makes Chat more diverse than PT counterparts? is there any misunderstanding?

---

> > ### Comment · Reviewer_L9QX · 2024-11-22
> >
> > (W-1-1): I understand the authors' response and acknowledge the difficulty of proving that the datasets in question are indeed part of the LLM's training data. However, even without definitive evidence, the paper fails to provide suggestive evidence to support this claim. While I am inclined to agree that a monoculture may exist in LLMs, the paper did not do a scientific demonstration and measurement of this phenomenon.
> >
> > Sorry that I didn't bring this up earlier, but the below paper might offer a method to provide a dataset is in training data.
> > Oren, Yonatan, Nicole Meister, Niladri Chatterji, Faisal Ladhak, and Tatsunori B. Hashimoto. "Proving test set contamination in black box language models." arXiv preprint arXiv:2310.17623 (2023).
> >
> > (W-1-2):
> > "goodreads data is a part of the training data as confirmed above". Are you able to confirm? In paper, the authors wrote "Goodreads reviews (Wan et al., 2019) (i.e., src)—very likely a portion of LLM training data (Achiam et al., 2023)".
> >
> > "It is hard to fathom that the overall (unknown) distribution of book reviews is this skewed, with next to no negative samples" Fundamentally, neither the authors nor I have any clear understanding of the scale of the unmeasured counterpart related to Goodreads reviews. However, it is the authors' responsibility to provide, at a minimum, suggestive evidence—rather than relying solely on rough reasoning—that (1) Goodreads data is included in the training data, and (2) it accurately reflects the overall distribution. I could not find such evidence in the paper.
> >
> > One possible scenario is that the unmeasured counterpart to Goodreads data is (1) entirely positive and (2) significantly larger—by an order of magnitude—than the Goodreads data. In this case, the overall distribution would effectively become overwhelmingly positive. While (2) seems quite plausible, I am uncertain whether and how (1) could occur. However, the authors have not ruled out this scenario, apart from expressing their belief that (1) is unlikely. Overall, I would have appreciated a more grounded and substantiated argument.
> >
> > This summarizes my overall feedback. While I am inclined to agree that a monoculture exists in LLMs, I find the demonstration and measurement of monoculture presented in this paper unfounded.
> >
> > The other responses addresses my questions well

---

> ### Author Response · Authors · 2024-11-22
> **Thank you for your feedback**
>
> 1. We are well aware of the existence of the work by Oren et al. However, as noted in their abstract: "the tendency for language models to memorize example order means that a contaminated language model will find certain canonical orderings to be much more likely than others. Our test flags potential contamination whenever the likelihood of a canonically ordered benchmark dataset is significantly higher than the likelihood after shuffling the examples.". Thus, this paper is also not useful in helping understand the training distribution for several reasons: (a) it works on understanding "memorization" -- which may not have happened at all (in fact, we don't see verbatim memorization in the generated outputs), (b) it is possible that the dataset originally was shuffled to begin with (their tests assume that the dataset was originally un-shuffled, and they compare this with a shuffled counterpart), and (c) their evaluation is done for models of size 1.4b, and there's no definitive guarantee that this is going to be reliable or useful for larger models. This is why we chose not to invest our limited rebuttal time in pursuing this rabbit hole.
>
> 2. It is completely perplexing for us that the reviewer thinks that GoodReads has a low probability of being included, when prior work shows far more rudimentary tabular datasets have been memorized (https://arxiv.org/pdf/2404.06209); none of the other reviewers share this concern and prior ``published'' works make similar (not the same) observations with NO knowledge of the training data (https://arxiv.org/pdf/2402.05070, https://arxiv.org/pdf/2303.17548, and https://aclanthology.org/2024.emnlp-main.244/).
>
> 3. Based on your feedback, we have considered a NEW dataset that is provably OUTSIDE the cutoff date for training these models; this means that your concern of contamination should now be resolved. We have run experiments, and shown that w.r.t this NEW dataset, monoculture exists. At this point, we fail to understand what remaining concerns the reviewer has. If the reviewer believes that this NEW dataset is also influenced somehow by the old unknown training distribution (without articulating how or why), then it's unclear what experimental evidence will satiate them -- their claim that the demonstration of monoculture being unfounded is both harsh and unreasonable. Particularly when the reviewer also acknowledges that it is virtually impossible to truly learn the training distribution, or its interplay on task performance. By this rationale, work of this nature can never be published unless large companies share their data composition, or truly open source models (like OLMo) are performant on a wide variety of tasks.
>
> 4. If everything else that we are saying is convincing, and you too likely believe that monoculture exists, why not reflect this in your score? It seems to us that the reviewer is ideologically in agreement with the phenomenon we present, appreciates the nuances in the experiments, and can't think of a better methodology than we have already presented. Yet the reviewer states that this work is not good enough?

---

> ### Author Response · Authors · 2024-11-22
> **Thanks for the great questions.**
>
> We would first like to shed light on the following differences:
>
> 1. Setting A: The traditional training ecosystem operates as follows: There is pre-training (the model obtained only after the step is denoted PT), followed by instruction fine-tuning (IFT) and preference tuning (or alignment). This final model is called Chat. The results outside the appendix are presented for models trained in this regime.
>
> 2. Setting B: However, for the models trained in the appendix, the pipeline is as follows: there is an additional round of supervised fine-tuning (SFT) done using the new dataset on both the PT model -- resulting in PT-SFT model, and on the Chat model resulting in the Chat-SFT model. In an ideal universe, we would have liked to add the new dataset to the PT phase for the Chat models too, but can't do so. So this is a coarse grained approximation of that setup.
>
> Having established this, please find our responses below:
>
> 1. In Setting A, we have consistently noted that in comparison to the source distribution, both PT + Chat models are less diverse. These are the results that are presented in the main body of the paper.
>
> 2. In Setting B (for figure 33a), we see that both (i) PT-SFT , and (ii) Chat-SFT models are less diverse compared to the source distribution: 3 colors in PT-SFT, Chat-SFT vs. 4 colors in source for prompt 1. In prompt 2 (figure 33b), we see that the PT-SFT model is less diverse than the source, but the Chat-SFT model is comparable.
>
> 3. Why is Chat-SFT more diverse than PT-SFT in figure 33b? Recall setting B: We perform additional fine-tuning on a previously aligned model to obtain the Chat-SFT model. We conjecture this to be the reason for increased diversity in prompt 2 (which explicitly requires the model to follow a particular persona in the instruction i.e., has a greater ``degree of instructability''). However, we would like to stress that 3 out of the 4 scenarios considered show the monoculture phenomenon clearly (reduced diversity compared to the source) in this non-standard training regime i.e., RLHF is likely to hurt diversity if it is the "last step" in model training (which is not the case for the Chat-SFT models).
>
> We hope this has clarified your concerns. Thank you for giving us this opportunity.

---

> ### Comment · Reviewer_L9QX · 2024-11-22
>
> Could the authors help me understand why "new, unseen dataset: We filter out books published after October 1, 2023 from the GoodReads dataset (May 2024 ver)" is "provably OUTSIDE the cutoff date for training these models"? Isn't this dataset consisting of books before October 1, 2023? Shouldn't be books after October 1, 2023 be provable "OUTSIDE the cutoff date"? Did I miss anything?

---

> > ### Author Response · Authors · 2024-11-22
> > **Apologies for our phrasing!**
> >
> > When we say we filter out books, we meant to say that we only consider books published after October 2023 and their corresponding reviews as the new data i.e., no books + reviews from before October 2023. Since the model's cut-off is October 2023, this is not in the training data.

---

> > > ### Author Response · Authors · 2024-11-22
> > > **other clarifications/questions?**
> > >
> > > Thank you so much for engaging— is there anything else that would be helpful to clarify or discuss about this issue?
> > >
> > > More generally, is there anything else that’s keeping your score from below a six? Is there a point you feel we haven’t addressed or a new concern you have? We’re happy to answer more questions or provide additional evidence if possible to reach this point. Please let us know.
> > >
> > > If you feel we have addressed your concerns, we would really appreciate you raising your score to a 6 or higher. Such a score is a signal to the AC that you’re not against the paper getting accepted, which we feel would accurately reflect the assessment if there are no remaining concerns. Of course, if this is not the case, we look forward to hearing your continued questions. Thank you!

---

> > > > ### Author Response · Authors · 2024-11-26
> > > > **We missed a comment!**
> > > >
> > > > *Comment: I remain concerned about the metric aspect due to the inaccuracy of the training distribution, which is unavoidable for most LLMs with unknown training data. In my view, the best way to address this concern is to conduct another experiment on OLMo (both pre-trained and aligned ones) with a task it can handle. So that this paper gives one measurement for monoculture from one experiment that has no concerns of inaccuracy.*
> > > >
> > > > **Response:** Thank you for your response, and our apologies for missing this point of yours.
> > > > To respond to this comment, we revisit the initial concerns that were laid out in the review. The reviewer initially questioned the validity of our experiments as we were unable to provide concrete proof that the models contained the GoodReads dataset. Upon revisiting this question, we are able to show that Goodreads is included in the training data. Work by Baack highlights that Llama v1 is trained on CommonCrawl [1]. Upon closer inspection (using the common crawl interface), one can see that the goodreads.com website and its associated URLs are part of this dataset. To verify this yourself, please visit https://index.commoncrawl.org/ and check for “goodreads.com” in the May/June checkpoint; we have also attached a screenshot in Appendix J (Figure 34). This suggests that this model includes book reviews from this dataset. While Baack notes that there is limited visibility into the training dataset for Llama 2, we can safely assume that it was built atop of the data collected for Llama 1, including CommonCrawl. Thus, we would like to stress that the original experiments in our paper and submission are valid (alleviating concern W1-1 raised). Thus, we strongly believe that performing the experiment on OLMo is unnecessary, and what we have done (both as part of the rebuttal and the main submission) using Llama is sufficient, and does not have any concern of inaccuracy.
> > > >
> > > > Completely resolving concern W1-2 is virtually impossible. Concretely, this would require us to show generative monoculture on a task that is influenced only by one portion of the training data, and nothing else. This alone is elusive. Further, extensive research has shown that the generative capabilities of these models emerge from the mixture of training data [2,3,4], so it is unclear if (a) we’d see powerful generative capabilities, let alone monoculture, in a setting where there is this “non-interference” property (which would also require us to train a model with a highly controlled data mixture), and (b) one can attribute the generation to an isolated sub-population in the training distribution.
> > > >
> > > > To summarize our conversation: recall that in your review, you acknowledged that the phenomenon we present is “subtle” and not immediately noticeable to all areas of society that warrants “further study” – we thank you for this assessment. As part of our response, we modified the training setup to include a “new dataset” that is outside the cut-off date for these models and show that monoculture exists even with full knowledge of the (particular) training distribution. We believe that this experiment (a) conclusively proves our claims, (b) is more detailed than previously published works that describe diversity issues in LLMs, and (c) has made our submission substantially stronger – thank you for the great comments! As part of experiments for other reviewers, we emphasize how the monoculture phenomenon exists despite model size scaling, and how naive mitigation strategies fail.
> > > >
> > > >
> > > > We really hope that you are positive about our work and its findings; the paper has been modified to reflect your suggestions. Should you believe that this phenomenon requires greater societal attention, and our findings and methodology is strong (as other reviewers point out), we humbly request you to raise your score to 6 or higher to send a very clear signal to the AC/SAC about your intentions in this competitive landscape.
> > > >
> > > >
> > > > Please let us know if this resolves your concerns. We are happy to continue to engage.
> > > >
> > > >
> > > >
> > > > References
> > > >
> > > > [1] https://facctconference.org/static/papers24/facct24-148.pdf
> > > >
> > > > [2] https://arxiv.org/abs/2402.09739
> > > >
> > > > [3] https://arxiv.org/abs/2405.14908
> > > >
> > > > [4] https://arxiv.org/abs/2406.11794

---

> > > > > ### Comment · Reviewer_L9QX · 2024-12-01
> > > > >
> > > > > Dear authors, thank you for your response. I have read it in details and will continue with my current rating.
> > > > >
> > > > > The largest concern for me remains the risk of inaccuracy for all concrete numbers of generative monoculture measured in this paper, due to inaccuracy of source distribution. The metric and measurements are important part of this paper. I find it difficult to trust the concrete generative monoculture measurements made in this paper, considering the authors take one dataset from the training data and measure the source distribution using this specific dataset. Not to say the current results are not helpful, but they would be more convincing if they are complemented with several measurements where the source distribution is accurate.

---

> ### Comment · Reviewer_L9QX · 2024-11-22
>
> Hey authors,
>
> thank you for taking the time to rebuttal. I have raised my rating because:
> (1) the authors included an additional experiment with an accurate source distribution, providing concrete evidence of generative monoculture. Combined with the main text's findings, despite uncertainties about the source distribution, I agree that the paper successfully demonstrates the phenomenon of generative monoculture.
>
> I remain concerned about the metric aspect due to the inaccuracy of the training distribution, which is unavoidable for most LLMs with unknown training data. In my view, the best way to address this concern is to conduct another experiment on OLMo (both pre-trained and aligned ones) with a task it can handle. So that this paper gives one measurement for monoculture from one experiment that has no concerns of inaccuracy.
>
> I agree that Oren et al. is not particularly helpful here, as confirming the dataset is part of the training data only addresses a small portion of the issue. The source distribution can still be inaccurate. Ideally, this could only be resolved if the full training data were known.

---

> ### Author Response · Authors · 2024-12-02
> **We do appreciate your engagement!**
>
> We thank the reviewer for engaging with us during the rebuttal. This ICLR, a common concern that was raised was the lack of reviewer engagement, so thank you for taking the time to respond to most of our comments. We did not want to bother you over the weekend, hence our delayed response.
>
> We understand your sentiment, but as stated several times during the rebuttal period, your concern is not one that can be addressed in any manner, let alone one that is reasonable within the time frame. To succinctly restate our response: even in scenarios where we are fully aware of the training distribution, it is impossible to determine if this particular distribution is fully responsible for the performance on a given task (unless the model is trained on one distribution to perform one task); thus using a random/representative sample of this distribution (that we can provably show was part of the training data) is the best approximation one can make. This is what we did in our experiments as part of the submission, and what we did for the experiment in the rebuttal as well. While the reviewer agrees that they believe the monoculture phenomenon, we are writing to understand if there’s any setting/experiment they are aware of that will convincingly respond to this concern? Note: Using OLMo is not useful, as stated in our rebuttal for 2 reasons: 1. It is not great for long form generation as needed in our experiments, and 2. We can't attribute generations to specific parts of the training data.

---

> > ### Comment · Reviewer_L9QX · 2024-12-02
> >
> > Dear authors,
> >
> > 1. I think the dataset used in experiments is neither "random" nor "representative". Could the authors highlight evidence that the source dataset is some form of random data from training data, or support that it is representative of training data? I acknowledge it is included or likely included in training data.
> >
> > 2. "it is impossible to determine if this particular distribution is fully responsible for the performance on a given task". I worry this statement adds additional risks to all findings from this paper to be honest. Other data will influence the performance of a task, too. It becomes another risk factor I think... but this is minor to my main concern.
> >
> > The only proposal I have is to work with OLMo with a task it can manage, and do an accurate generative monoculture measurement there. Or other ways to establish one more accurate generative monoculture measurement, with reasonable evidence to support the (approximate) accuracy of source distribution.

---

> > > ### Author Response · Authors · 2024-12-02
> > > **Thanks for the response, again!**
> > >
> > > 1. Neither of us can prove/disprove the randomness given that this is proprietary information, but it is one of the few large available datasets containing book reviews, and is proven to exist in the training data. We struggle to see how using this as a source for showing book-review generations can be completely unrepresentative.
> > >
> > > 2. We understand the desire for a perfect monoculture measurement, but we really believe that our method is the best possible given the constraints we have laid on in this conversation. Using OLMo, a model that's not attuned for long-form generation task to show a phenomenon that is termed "generative" monoculture is not equipped to show the phenomenon and would not actually abate your concerns.  We believe that this additional experiment would provide no conclusive value beyond what we have provided already.
> > >
> > > We have performed many experiments (all of the ones that were possible to do and we felt would strengthen the paper) during the writing of the paper and the rebuttal period. Without a concrete alternative/experimental procedure for how to address this final concern, we hope that the evidence we have presented so far will suffice for you to move towards recommending acceptance. While we cannot do any more experiments to update the paper during the review period, if there is something you would like us to include conditional on acceptance, we are happy to do so.
> > >
> > > Thank you so much again for your continued engagement.

---

### Official Review · Reviewer_N2Ye · 2024-10-30

**Soundness:** 3
**Presentation:** 3
**Contribution:** 3
**Rating:** 6
**Confidence:** 4

**Summary:**

This paper introduces the concept of “generative monoculture”, which refers to the phenomena that an LLM's output distribution is less diverse than the original input (data) distribution.
Generative monoculture is investigated on two tasks: generating book reviews, and generating coding solutions, across multiple LLMs.
The monoculture is measured with a number of metrics (depending on the task): distribution of the mean, entropy and standard deviation, and mean pairwise similarity.
The authors find that monoculture exists across the LLMs that are investigated. The authors propose several mitigations, such as changing the sampling parameters, or diversifying the prompts, but this has little impact.

**Strengths:**

* **Interesting quantification of generic monoculture:** As the authors also mention in their related work section, the idea that current alignment practices hurt diversity is not necessarily new. However, this work presents a new way of measuring this, which gives additional insights in the output of (aligned) LLMs.
* **Considerations (pros and cons) of approach are clearly presented, and largely make sense:** The authors clearly mention their considerations for adopting their chosen methodology, in a space where they sometimes had to make some shortcuts or assumptions. For example regarding their data selection and their used metrics.
* **Authors present negative results, that are insightful:** The presented mitigation strategies do not really help, but this is an insightful finding for the community.

**Weaknesses:**

* **Investigated datasets are not very large.** This holds especially for the code solutions, where the data is limited to a subset of 100 easy solutions. Although this gives a first impression, I wonder how results hold over larger data samples, and especially when harder problems are included as well.
* **Coding solutions are checked by GPT-3.5.** The authors give some details in the appendix of the autojudge, but it is not entirely clear to me how quality is ensured. After all, GPT-3.5 needs to check the results of a stronger model (GPT-4) and, as the authors also mention elsewhere in the paper, LLMs tend to be biased towards their own responses.

**Questions:**

* Can the authors explain what they mean on line 309: “We instantiated this by generating k samples [...], and verifying that at least 20 of them were correct.” Does this mean 20 samples were checked, and 80 were not?
* Can the authors provide a reference that shows that humans prefer positive reviews, to support take-away 2?

---

> ### Author Response · Authors · 2024-11-20
> **Thank you for your feedback!**
>
> We thank the reviewer for the comments and questions. Please find our response below.
>
> **Generation and correctness evaluation of code solutions.** Let us clarify–we kept generating solutions and measuring their correctness, until at some point all problems reached at least 20 correct solutions. We then stopped. That is why we ended up generating 100 solutions per problem for GPT-4 and 200 for Claude-3–essentially, GPT-4 is better at generating correct solutions and we stopped early for it while we had to continue generating more for Claude-3. Comparing Fig. 5a with Fig. 26 in Appendix, we can see that GPT-4 does achieve higher accuracy in general compared to Claude-3.
>
> **Size of the Code dataset.**
> a) The reason we are limited to working with “easy” problems is the following: the model generated solutions on harder questions achieved extremely low correctness/accuracy, while for the purpose of our study (on the other attributes) we focus on only the correct solutions (lack of diversity is not as important when the generated solutions themselves are incorrect) and we need at least 20 correct solutions. By our procedure (described in “Generation and correctness evaluation of code solutions”), that means we needed to generate a large number of solutions before having just a few that are correct; this experiment was very expensive (because of the large number of API calls needed). This is the inherent difficulty posed by the limited capability of the models, and we do not have better solutions before modes become more inexpensive, and get more capable at generating correct code.
>
> b) Regarding the set of 100 problems, we wanted to clarify that this means we generated and autojudged 10,000 solutions (100 x 100)  in total by GPT-4 (and 20,000 solutions in total by Claude-3) on these 100 problems (described in “Generation and correctness evaluation of code solutions”). Moreover, the results on 100 problems already revealed pretty significant trends. It is true that we can aim for a larger dataset, but within a limited budget (these experiments already cost us more than $2,000 in invoking the APIs) and out of consideration for efficient and environmental-friendly experimentation, we strongly believe 100 is a reasonable number.
>
> **Coding attributes provided by GPT-3.5.** We agree with the reviewer that the quality needs to be ensured. Towards this purpose, the several authors on this paper have (independently) manually validated the quality of the responses given by GPT-3.5–-detailed procedures and results can be found in Appendix E.3 and Table 3, referenced at line 313 in the main paper. The scores in Table 3 show that the extracted results given by GPT-3.5 are highly accurate and reliable. In addition to the attributes extracted by GPT-3.5, we also included attributes obtained from other means to serve as a supplement wherever possible, e.g., for measuring “efficiency”, in addition to the “complexity” results provided by GPT-3.5, we also measured the runtime and memory usage as a support. We hope these together increase the credibility of the results.
>
> **Humans prefer positive reviews.** We are happy to provide some references [1,2,3]. We additionally want to point out that, what we mean here is not merely human preference *in general*, but human preference when specifically infused in preference tuning (like RLHF). It’s probably more accurate to put it another way: “preference tuning enforces the preference for positivity, correctness, and efficiency for a language model assistant.“
>
> **References**
>
> [1] Dodds, Peter Sheridan, et al. "Human language reveals a universal positivity bias." Proceedings of the national academy of sciences 112.8 (2015): 2389-2394.
>
> [2] Augustine, Adam A., Matthias R. Mehl, and Randy J. Larsen. "A positivity bias in written and spoken English and its moderation by personality and gender." Social Psychological and Personality Science 2.5 (2011): 508-515.
>
> [3] Boucher, Jerry, and Charles E. Osgood. "The pollyanna hypothesis." Journal of verbal learning and verbal behavior 8.1 (1969): 1-8.

---

> > ### Comment · Reviewer_N2Ye · 2024-11-20
> > **Thank you for your response!**
> >
> > Thank you for your response! This clarifies my questions -- it would be great if you could update the paper accordingly.

---

> > > ### Author Response · Authors · 2024-11-21
> > > **Thank you!**
> > >
> > > Thank you! We will work on updating the draft with this content. Would you consider raising your score in support of the work?

---

> > > > ### Author Response · Authors · 2024-11-22
> > > > **Happy to engage!**
> > > >
> > > > Thank you for the wonderful questions; as we near the end of the rebuttal period, and as we work on updating the draft, we are happy to engage during the rebuttal period and clarify any concerns! Please do consider our earlier request!

---

> > > > > ### Comment · Reviewer_N2Ye · 2024-11-23
> > > > >
> > > > > Thanks for your response. I appreciate the clarification, and although it clarifies the method, I don't think it fundamentally changes the overall quality of the paper. Having read the other reviews, and the discussions that followed, I think a score of 6 reflects my overall assessment of the paper well.
> > > > >
> > > > > Thank you for your engagement throughout the rebuttal period!

---

### Official Review · Reviewer_rybD · 2024-11-03

**Soundness:** 3
**Presentation:** 3
**Contribution:** 3
**Rating:** 8
**Confidence:** 4

**Summary:**

The paper defines and documents the problem of “generative monoculture” in LLMs – that is, the situation in which LLMs produce outputs that are significantly less diverse that what was present in their input training data. The authors focus on diversity over task-specific and intuitive metrics – specifically, the sentiment of book reviews or the algorithms employed in generated code. This is in contrast to measuring over some more systematic but arguably less informative metric such as lexical distributions. The authors find that, across models, the problem of generative monoculture is present and argue that it seems to get worse for models which are “aligned” to human preferences via RLHF.

I really like this paper. It's a nice, intuitive idea that deserves to be highlighted. I think the study was executed well for the most part but would have preferred to see some human evaluations, rather than solely automatic ones. But I think despite this, it warrants publication in the current form.

**Strengths:**

The paper formalizes the idea of “monoculture”. This idea isn’t wildly novel–it’s intuitive and consistent with other similar ideas such as mode collapse–but to my knowledge there isn’t a clean documentation of it and thus the paper has value in being an official cite for this phenomenon

The authors focus on measuring monoculture using task-specific notions of salient attributes (e.g., sentiment in book reviews, algorithms in code) which differs meaningfully from measures that use e.g., vocabulary of generations. I think this distinction is meaningful as it's a better measure of the type of distribution shift that will matter in practice if GenAI is widely deployed.

**Weaknesses:**

My primary concern is that the evaluation focuses entirely on automatic metrics. Granted, there are many metrics that the authors use, and they are somewhat diverse. Still, many of the metrics rely on using LLMs themselves (mostly GPT 3.5) to evaluate LLM output. There is something circular (though hard to articulate) about doing this especially given the premise of the paper itself. That is: if we assume LLMs are not good at generating diverse outputs, might we also worry that they aren’t good at recognizing such diversity? Or, said differently, why are we confident that the collapse is due to actual differences in what the LLMs produce, and not differences in what the LLM evaluator can detect?

To address this, I think the paper should include some evaluations which are determined entirely by human judgments. I.e., ask humans to rate the sentiment rather than asking LLMs to do so, ask humans to evaluate the big-O complexity rather than having GPT do it, etc. I would be dramatically more convinced if the conclusions held up under human eval, rather than just automatic eval. But, admittedly, I do expect the result to hold even if all evaluations were switch to human eval.

**Questions:**

None beyond what was raised in weaknesses

---

> ### Author Response · Authors · 2024-11-20
> **Thank you for your feedback!**
>
> Thank you for your review. We offer more discussion with respect to the concerns raised.
>
> First, we agree with the reviewer that human evaluation could further help increase the credibility of evaluations and we have in fact made efforts towards this. Several authors on this paper have (independently) manually validated the quality of the responses given by GPT-3.5–-detailed procedures and results can be found in Appendix E.3 and Table 3, referenced at line 313 in the main paper; this was provided as part of the original submission. The scores in Table 3 show that the extracted results given by GPT-3.5 are highly accurate and reliable.
>
> More broadly, the reviewer raised a valid point regarding the discriminative capability of LLMs. The question itself is interesting and has not been well studied yet. But there is evidence supporting contrasting sides of the argument [1,2], with [1] claiming that LLMs are better discriminators while [2] criticizing the understanding of LLMs. But, whether LLMs can be good discriminators or not is orthogonal to the message of our work; the conclusions we derive with respect to the diversity of the generations shall not hinge on the answer to this question. This is because 1) we have decomposed the discrimination task to a simpler form (e.g., sentiment classification, an easy task with human provided labels for ground truth exist for many datasets, and classifiers are trained for this very task) and 2) for these tasks, we have made sure to adopt well-trained models that are commonly used.
>
> Thank you very much and please let us know if you have follow up questions or concerns.
>
> **References**
>
> [1] Are Language Models Better at Generating Answers or Validating
> Solutions? https://openreview.net/pdf/d3f724811cb971ff60cd45abfc51bd92d2dd7602.pdf
>
> [2] West, Peter, et al. "THE GENERATIVE AI PARADOX:“What It Can Create, It May Not Understand”." The Twelfth International Conference on Learning Representations. 2023.

---

> > ### Author Response · Authors · 2024-11-22
> > **Happy to engage!**
> >
> > Thank you for the wonderful questions; we are happy to engage during the rebuttal period and clarify any concerns! Thank you for your positive encouragement and feedback for our work!

---

### Official Review · Reviewer_dgPh · 2024-11-04

**Soundness:** 2
**Presentation:** 3
**Contribution:** 4
**Rating:** 6
**Confidence:** 5

**Summary:**

The main idea of this paper can be summarized in the following toy example: if 90% of humans say that chocolate is tasty, should language models *always* describe chocolate as tasty, or should they aim to somehow reflect the diversity in human opinions, and occasional refer to chocolate as bad-tasting?

The authors apply this thought experiment to two domains---book reviews and code implementation---and analyze the various ways in which language models enforce a monoculture by failing to model the diversity present in actual human-written content. Adding more entropy into the decoding pipeline helps a bit, but does not meaningfully bring diversity to human levels.

The authors include an interesting discussion of the scenarios in which diversity is desirable--for example, generated book reviews ought to reflect the range of opinions real reader mights have on said books---and the scenarious where diversity might be less important---for example, it is more important for generated code to be correct than to be diverse.

**Strengths:**

The main idea of this paper is very interesting, and I am glad the authors have done this exploration. The authors have done a good job of discussing nuances around the merits of diversity, and I appreciate their selection of two complementary domains where the value of having diversity is quite different.

**Weaknesses:**

## Primary weakness - incomplete description of methodology
Unfortunately, it is not possible to assess this paper as it was submitted because crucial information required to understand and reproduce the methodology is purported to be in the appendix, but no appendix was included in the submission. Since the paper is incomplete, there is no choice but to give a score of 1 (strong reject). Despite this, I have tried to leave some constructive feedback below for the authors.

## LLMs for attribute extraction
Line 163 states that "care must be taken to use LLMs for attribute extraction, as they are known to be biased towards their own responses." It seems like this care was taken for the sentiment classifier (the paper notes the classifier's accuracy on SST-2 and it's widespread usage). However, the same care does not seem to have been taken for the "efficiency" attribute, which uses gpt-3.5 to assess runtime efficiency of generated code. It is unclear why a reader should trust that gpt-3.5 is efficient and unbiased at this task.

## Takeaway #2

Line 393 states that humans "largely prefer text with positive sentiment." Is this actually true? In a 5 minute literature review, I found several papers such as the two listed below which suggest a much more complex story. Needless to say, statements such as this one should not be made without citation.

Sangwon Park, Juan L. Nicolau. Asymmetric effects of online consumer reviews, ISSN 0160-7383, (https://www.sciencedirect.com/science/article/pii/S0160738314001273).

Lotte M. Willemsen, Peter C. Neijens, Fred Bronner, Jan A. de Ridder. “Highly Recommended!” The Content Characteristics and Perceived Usefulness of Online Consumer Reviews, (https://academic.oup.com/jcmc/article/17/1/19/4067647).

## Takeaway #3

Takeaway #3 claims RLHF hurts diversity more than any of the other factors did. This claim would be much stronger if it had been backed by experiments with more than just one pair of models. I would like to see additional experiments with other model pairs (for example instruction-tuned and RLHF'ed OLMO). Otherwise, a caveat should be added to this claim about the limitation of the result.

## Nitpicks

Here are a few nitpicks (which wouldn't much affect my overall assessment).
1. I am not a fan of the phrase "human-generated." It is extremely atypical to talk about a human generating a book review or generating some code (unless the speaker wants to imply the human is using genAI tools); rather, in common parlance, humans **write** book reviews and **write** code. I suggest replacing instances of "human-generated" with "human-written."
2. I think the paper would be more engaging to read if the description of the attributes of interest was moved before the section on metric calculation (Section 3.3). Section 3.3 felt out of place to read when I didn't yet know what exactly the attributes were for each domain.
3. Figure 4 has too much information in it, which impeded communication. It would be more effective to break this into multiple figures each with their own caption. In particular, the (a), (b) and (c) in the middle bar graph are especially confusing since (a), (b) and (c) are also used to refer to subfigures. Can you instead put shorthands for the actual names of the models? Also, for the topic model I am confused why the word 'novels' occurs in two groups.
4. For Figure 5, an easier-to-understand x-axis label for the top-right plot would be "Self-similarity" rather than "Similarity."
5. One additional paper citation for the first paragraph of your Related Work section ("Forcing Diffuse Distributions out of Language Models" https://arxiv.org/abs/2404.10859).
6. Many of the references are formatted incorrectly with missing capitalization in the paper titles.

**Questions:**

1. What was `{person}` replaced with in the prompt?
2. I do not understand the method used to generate correct solutions for the coding task. Specifically, I did not understand this sentence (line 309): We instantiated this by generating k samples (100 for GPT-4 and 200 for Claude-3), and verifying that at least 20 of them were correct." What did you do if it wasn't the case that 20 were correct? Does your sampling method here ensure then that "correctness" will never fall below 20%?

---

> ### Author Response · Authors · 2024-11-20
> **Thank you for your feedback!**
>
> We thank the reviewer for the comments and questions and would like to offer some clarifications and hope that addresses some of the reviewer’s concerns.
>
> **Incomplete submission (appendix).** Our appendix can be found in the supplementary materials. This was submitted along with the original paper, in accordance with the guidelines specified where the appendix could be part of the main paper, or be included with the supplementary material. We hope that since this was the main reason the reviewer cited for giving us a score of 1, the reviewer will raise their score.
>
> **LLMs for attribute extraction.** The reviewer made a good point in that we should use care in attribute extraction. We hold the same belief that the extraction functions need to be “efficient, accurate, and reproducible” (line 161). Towards this purpose, the several authors on this paper have (independently) manually validated the quality of the responses given by GPT-3.5, not only for the “efficiency” attribute but also other attributes like “description” and more–-detailed procedures and results can be found in Appendix E.3 and Table 3, referenced at line 313 in main paper. The scores in Table 3 show that the extracted results given by GPT-3.5 are indeed highly accurate and reliable. In addition to extracted “efficiency”, we also measured the actual runtime and memory usage of the generated code (i.e., using system profiling tools), to complement the evaluation on “efficiency”. We hope these together increase the credibility of the results.
>
>
> **Human preference.**  We want to clarify that what we meant was not human preference in general, but the human preference *specifically infused in preference tuning (like RLHF)*. It’s probably more accurate to put it another way: “preference tuning enforces the preference for positivity, correctness, and efficiency for a language model assistant.” We are happy to provide citations for the idea that humans prefer positive text [1,2,3]: the seminal work of Dodds et al. [1] states that “Using human evaluation of 100,000 words spread across 24 corpora in 10 languages diverse in origin and culture, we present evidence of a deep imprint of human sociality in language, observing that (i) the words of natural human language possess a universal positivity bias, (ii) the estimated emotional content of words is consistent between languages under translation, and (iii) this positivity bias is strongly independent of frequency of word use.”  Similarly, the work of Augustine et al. [2] states that “The human tendency to use positive words ("adorable") more often than negative words ("dreadful") is called the linguistic positivity bias. We find evidence for this bias in two studies of word use, one based on written corpora and another based on naturalistic speech samples.“
>
>
> **Nitpicks.** Those are all great suggestions for improving the clarity and readability of the paper, the presentation of the figures and formatting of the references. We will incorporate them in our revision.
>
> **Topics.** For the clarification on topic models: each topic id corresponds to a topic that’s characterized by a list of keywords (we showed 2); together, these keywords are most representative of the topic and can be thought of as a concrete way to understand the embedding of the topic. Assignment of topic is not done by matching with these keywords but by comparing the text embeddings. So it’s normal that “novel” can occur in multiple topics, as these topics could concern just different genres of novels.
>
> **{person}.** The list can be found in Appendix B.2. The list of names are as follows: Trevor Noah, Janelle Monáe, Yuval Noah Harari, Serena Williams, Reshma Saujani, Neil deGrasse Tyson, Margaret Atwood, David Attenborough, Malala Yousafzai, Jordan Peele.
>
> **Generating code solutions.** To clarify our process: we kept generating solutions and measuring their correctness, until at some point all problems reached at least 20 correct solutions; we stopped then. That is why we ended up generating 100 solutions per problem for GPT-4 and 200 for Claude-3–essentially, GPT-4 is better at generating correct solutions and we stopped early for it while we had to continue generating more for Claude-3. Comparing Fig. 5a with Fig. 26 in Appendix, we can see that GPT-4 does achieve higher accuracy in general compared to Claude-3.

---

> ### Comment · Reviewer_dgPh · 2024-11-20
> **Raised review to a 5**
>
> Thank you for your detailed response.
>
> I would not be opposed to this paper getting accepted, but I do think it could benefit from another round of revisions.

---

> > ### Author Response · Authors · 2024-11-20
> > **Thank you for changing your score!**
> >
> > As you are aware, a score of 6 indicates that the paper is marginally above the bar for acceptance and would reflect your sentiment more accurately, if you feel like the paper can be accepted. However, your score is currently at 5 (marginally below the bar for acceptance). What additional revisions can we perform to help you further change your score? We are happy to engage during this rebuttal period, and thank you for the opportunity to strengthen our work.

---

> > > ### Author Response · Authors · 2024-11-22
> > > **Happy to engage!**
> > >
> > > Thank you for the wonderful questions; we are happy to engage during the rebuttal period and clarify any concerns! Please let us know if we can provide any other information to help with a score change.

---

> > > > ### Author Response · Authors · 2024-11-22
> > > > **could you clarify your other concerns?**
> > > >
> > > > We hope this message finds you well. You mentioned that you felt the paper could go through another round of revisions-- could you please clarify what remaining concerns you have that is keeping your score below a 6? Is there a point you feel we haven’t addressed or a new concern you have? We’re happy to answer more questions provide additional evidence if possible to reach this point. Please let us know. We are uploading the revised version of the paper today which reflects your feedback, the changes will be in blue.
> > > >
> > > > If you feel we have addressed your concerns, we would really appreciate you raising your score to a 6 or higher. Such a score is a signal to the AC that you’re not against the paper getting accepted (which you did state in a previous response), which we feel would accurately reflect the assessment if there are no remaining concerns. Of course, if this is not the case, we look forward to hearing your continued questions. Thank you!

---

> > > > > ### Author Response · Authors · 2024-11-26
> > > > > **Thanks again!**
> > > > >
> > > > > Thank you for raising your score from a 1 to a 5 after acknowledging that the supplementary material was indeed submitted as part of our original submission. As we near the end of the rebuttal period, we are writing to request if you have any other concerns we may be able to clarify.
> > > > >
> > > > >
> > > > > Recall that in your review, you found our main idea “interesting” and were “glad” with the level of the experiments performed. As part of the rebuttal period, we have addressed many of the nits you have raised through changes to the text in the paper, especially changing all “human-generated” to “human-written” and providing more references. We have provided more details about our experimental methodology, and provided citations regarding human preferences and clarified the writing in our draft. As part of additional experiments, we show that monoculture persists despite model scaling, and in scenarios where we have perfect knowledge of the training distribution.
> > > > >
> > > > >
> > > > > We hope you are still positive about our work; the paper has been revised as per your specifications. It has indeed become stronger! Should you believe that the findings are interesting and publishable, we request that you consider raising your score to 6 or higher, to send a clear signal to the AC/SAC about your interest in this competitive landscape. We will gladly make all the editorial changes you have suggested should the paper be accepted for publication. This can not happen with your support!

---

> > > > > > ### Comment · Reviewer_dgPh · 2024-12-03
> > > > > > **Raised my score to 6**
> > > > > >
> > > > > > Apologies for not engaging more during the discussion period.
> > > > > >
> > > > > > I have looked more closely at your additional experiments and other revisions, and I have decided to raise my score to 6. As I mentioned in my previous comment, I don't think there is any harm of this paper being published at ICLR, and it might be of interest to some members of the community.

---

### Official Review · Reviewer_SPD6 · 2024-11-04

**Soundness:** 2
**Presentation:** 2
**Contribution:** 2
**Rating:** 5
**Confidence:** 4

**Summary:**

This paper addresses the critical issue of "generative monoculture" in large language models (LLMs), where output diversity narrows compared to training data. This is particularly concerning as LLMs are increasingly used in diverse applications like product reviews, sentiment analysis, and scholarly summarization. The study focuses on book reviews and code solutions to demonstrate this phenomenon and tests various mitigation strategies, such as temperature adjustment and prompt engineering, though with limited success.

**Strengths:**

This paper highlights a critical issue in LLMs around narrowing of output diversity compared to the training data. The paper addresses an important problem esp when LLMs are being increasingly applied in diverse fields such as automated product reviews, sentiment analysis, scholarly paper summarization etc. The paper demonstrates the prevalence of narrowing of output diversity, which they refer to as 'generative monoculture'. They consider book reviews and code solutions as two primary use cases to study the narrowing phenomenon.
The paper tests various methods to mitigate 'monoculture', including temperature adjustment and prompting strategies.

The paper provides a well-structured methodology for measuring generative monoculture, including diverse metrics like entropy, mean pairwise similarity and other such metrics.

The paper discusses the impact of narrowing of output diversity on the societal and code security aspects. These discussions strengthens the paper.

**Weaknesses:**

While the paper tests various methods to mitigate 'monoculture', including temperature adjustment and prompting strategies, the attempted countermeasures showed limited efficacy in mitigating narrowing of output diversity. This warrants more experimentation and ideation. I would also think use cases/tasks other than book reviews and code generation should be investigated to test the generalizability of the method. Dialogue / chat bot as an application may be an important area to test these methods on. The inability to redefine the alignment process limits the degrees of freedom with which one could operate. I would have liked to see a bit broader scope that covers alignments and comparing the dispersions to alignment data rather than the training data of CPT.

Rich getting richer / Echo Chamber / Feedback loops are commonly studied phenomenon in recommender system literature. Methods like Bandits and Exploration/Exploitation are commonly used to mitigate the homogeneity of recommended items over time. Authors have an opportunity to connect to this rich work and draw inspiration from this field.

**Questions:**

I would like authors to comment on feedback loop effects. How do they anticipate monoculture evolving as LLMs are increasingly used to generate their own training data in feedback loops ? Do they think this will further exacerbate the monoculture effect. What strategies and/or guardrails do the authors suggest to mitigate this very long term "model collapse" situation ?

Authors could also further comment on composition of training data. Would increased diversity play a role in mitigating feedback loops and how can LLM training incorporate such diversity.

Lastly, what role model size plays ? Have authors ablated along model size/complexity dimension ?

---

> ### Author Response · Authors · 2024-11-20
> **Thank you for your feedback! (part 1)**
>
> We thank the reviewer for their thoughtful comments.
>
> ***Weaknesses***
>
> We briefly respond to the weaknesses mentioned by the reviewer. Overall, we agree that these are fascinating questions.
>
> With regards to weakness 1+3, we would like to stress that our primary contribution in this work is defining generative monoculture, and providing a methodology to measure it. While changing temperature and prompting strategies were not effective countermeasures to deter generative monoculture, these are important negative results as these are methods that are often understood to promote diversity and showcasing their ineffectiveness is important. Our work shows that the alignment process might be one cause for the monoculture phenomenon; however, designing and evaluating solutions in the alignment phase for popular models are both non trivial and expensive. One possible approach would be to consider alignment approaches such as DPO [3] which introduce a flavor of supervision, or perform alignment using a “mixture of reward models” with diverse preferences. However, it is unclear what implications these solutions may have on downstream effects, such as output saliency, coherence, correctness and safety. Exploring this will be interesting future research.
>
> For weakness 2, we agree that there are several cases where monoculture can be extended, such as studying how political preferences between train and model-generated data change. However, we believe that the two case studies we have picked are already representative. Extending monoculture evaluation to chatbots is unclear: what would the source distribution be? Measuring diversity change in this scenario would also require the nature and turn of dialogues to be the same as in this source distribution (for a fair comparison); this makes the scenario extremely convolved and hard to study.
>
> For weakness 4, we will certainly study the literature associated with bandits and reinforcement learning, and make appropriate connections in our work. Thank you for the pointers!
>
>
> **Q1 and Q2. Feedback Loops/Model Collapse and Monoculture**:
> a) *Overview and comparison*: First, we quickly remind the reviewer that the phenomenon that we demonstrate, generative monoculture, is separate from that of model collapse. Generative monoculture demonstrates that LLMs trained entirely on human-generated data reduce diversity/do not reflect the true range of information in their output. Model collapse shows how LLMs trained on *their own generated data* degrade in performance over time [1,2]. One can think of monoculture as a special case of collapse, where degradation is akin to progressive output diversity reduction.
>
> b) **(Q2) Strategies preventing model collapse, and its relation to our work:**
> Research on the (separate) phenomenon of model collapse has suggested that access to the original data is crucial [2], and noted that random sampling could serve to mitigate the issue [1].
> Importantly, there are different conclusions about these two related, but separate phenomena. We show that:
> Even LLMs trained on human (not model-generated) data have a narrower distribution than their training data. Despite the comments in [2] regarding the importance of original data–here, even with access to the original data, the issue of losing information in the true distribution still exists.
> Tuning sampling parameters cannot effectively mitigate the issue, contrary to what is suggested in work related to collapse [1].
>
> Together, these suggest that in generative monoculture, the loss of diversity from training data to model responses is complicated. Results (and findings) from work on feedback loops and model collapse may not directly transfer here, and not all solutions/mitigations for model collapse may not work effectively in terms of mitigating generative monoculture. As stated earlier, we regard generative monoculture as a close but distinct topic to feedback loop / model collapse that deserves separate attention (we have also involved some other discussions at the end of Sec 7).
>
> Regarding mitigating generative monoculture, we look forward to this in future work, but reiterate some preliminary thoughts here. Our analysis pointed out that generative monoculture is more of a concern for fine-tuned models (both instruction fine-tuned and RLHF fine-tuned, and more pronounced on the latter) and less of an issue for pre-trained models. Thus we would suggest making improvements to the fine-tuning *data* or the preference tuning *approach* to explicitly encouraging favoring diversity. One way to realize this is, instead of strictly optimizing the score of a single reward model, we optimize against a mixture of reward models with different preferences (look at response to weakness 1+3). Thus the diverse answers all have their chance to be promoted, instead of only one answer.

---

> > ### Author Response · Authors · 2024-11-20
> > **part 2**
> >
> > c)  **(Q1) Feedback loops exacerbating monoculture:**  If we think of generative monoculture more generally as  reduced diversity in output compared with the source (not necessarily human input as we studied), then feedback loops can be understood as a composed sequence of generative monoculture, and reduced diversity in the training set (from model generations that are less diverse than human data) will likely lead to less and less diverse generations over time (assuming model generated outputs are used as input for subsequent training iterations, without including the human-created data).
> >
> > **Q3. Would increased diversity mitigate the issue and what’s the approach?** [1] suggests that increasing diversity by training on generated data obtained by random sampling instead of deterministic beam search would lead to more stable distribution under iterations of feedback loop. In our case, where we are more concerned about the step 1 from real training data to the 1st iteration of generated data, it’s unclear what’s the cause of the loss of information; we have hypothesized RLHF as one cause, but there might be others. One would need to investigate the adopted training/fine-tuning approach to understand the extent and direction of this happening in pre-training and different fine-tuning, and make changes to training objectives/techniques. Based on our understanding, increasing diversity will probably help (as expected), but is likely not sufficient on its own: as we can expect, the training set of GPT-4 and Claude must be huge and diverse, yet our experiments show that generative monoculture is still severe in them, indicating that there are causes/issues beyond data.
> >
> > **Q4. Influence of model size.** We conducted experiments to understand the pervasiveness of monoculture as a function of model size. To this end, we conducted the sentiment analysis study on Llama 70b (instruction fine-tuned + aligned); we used the quantized model for faster inference times given the narrow window of the rebuttal period. Recall that associated experiments in the draft were performed on LLama 13b. Our results show that while this larger model has “greater diversity” than the smaller model (potentially due to an increase in the training dataset used to train this model), the phenomenon of monoculture (i.e., decreased diversity compared to the source reviews) remains. We have added the figure in Appendix G for your review.
> >
> >
> > References
> > [1] Taori, Rohan, and Tatsunori Hashimoto. "Data feedback loops: Model-driven amplification of dataset biases." International Conference on Machine Learning. PMLR, 2023.
> > [2] Shumailov, Ilia, et al. "The curse of recursion: Training on generated data makes models forget." arXiv preprint arXiv:2305.17493 (2023).
> > [3] Direct Preference Optimization: Your Language Model is Secretly a Reward Model  Rafael Rafailov, Archit Sharma, Eric Mitchell, Stefano Ermon, Christopher D. Manning, Chelsea Finn
> >
> > We hope that we have satisfactorily responded to your queries, and you would consider raising your score for this work.

---

> > > ### Author Response · Authors · 2024-11-22
> > > **Happy to engage!**
> > >
> > > Thank you for the wonderful questions; we are happy to engage during the rebuttal period and clarify any concerns!

---

> > > > ### Author Response · Authors · 2024-11-26
> > > > **Thanks again!**
> > > >
> > > > As we near the end of the rebuttal period, we are writing to you to request if you have any other further clarifications w.r.t our work.
> > > >
> > > > Recall that in your review, you found our observation “interesting”, found our methodology “well structured”, and found our application scenarios “strong”. In response to your concerns, we clarified the relationship between monoculture and model collapse (initial discussion was already present as part of our submission), and discussed how solutions for model collapse may not directly translate over to monoculture. We conducted additional experiments using a larger model (Llama 70b) and showed that the phenomenon persists. In response to other reviewers’ concerns, we performed additional experiments to highlight the validity of our method (including our data selection strategies) and showed that in settings with full knowledge of the training distribution (which is virtually impossible for most models), monoculture is still prevalent.
> > > >
> > > > We hope you are still positive about our work. Should you believe that the findings are interesting and publishable, we request that you consider raising your score to 6 or higher, to send a clear signal to the AC/SAC about your interest in this competitive landscape.

---

> > > > > ### Author Response · Authors · 2024-12-03
> > > > > **Thank you again for your comments, is there anything else you need to raise your score?**
> > > > >
> > > > > We hope this message finds you well. Since this is the last day for the rebuttal, we are writing to inquire if there’s any other information you need from us to consider increasing your score; we are happy to engage.
> > > > >
> > > > > As requested, we have added more information beyond what was in our submitted draft to compare our work with model collapse, and reflected this in the revised draft. We also conducted experiments on larger models (llama-70b) and show that the monoculture phenomenon exists. We thank you again, for your feedback. It has helped improve our work.

---

### Author Response · Authors · 2024-11-22
**Thank you, reviewers, for your comments!**

We thank all of the reviewers for their time offering feedback on our work. We are happy to see that the reviewers thought that the phenomenon of generative monoculture is a critical issue that could have a large impact on society, and largely thought that our presentation of the phenomenon was well-executed. We also appreciated that a reviewer pointed out that our highlighting of negative results at ameliorating monoculture from temperature and prompting strategies was helpful to the community. We have revised the draft to incorporate comments below.

***Human Validation of LLM Evaluation*** The most common comment brought up by the reviewers was a desire for some human evaluation of the coding tasks. We would like to stress that evidence of this was already presented as part of our submission; Appendix E.4 had results showing how members of the research team have independently evaluated the LLM code description, time/space complexity, and algorithm tagging evaluations, and found them to be quite accurate. The protocol they followed was checking a random sample of LLM evaluation of code samples, and rated them on a scale from 1-3: 1 being completely incorrect and 3 being completely correct. All tasks received an average score of 2.5 or over, and 5 out of 9 tasks had an average score of 2.9 or over. During the rebuttal period, one other member of the study team carried out the same manual annotation task, and came to similar conclusions. Based on this assessment, we feel comfortable with the results obtained from LLM-generated code reviews, and hope the reviewers agree. More so, LLM evaluation and self-evaluation is quite standard and there is evidence to suggest that it is effective [1]. However, we appreciate this concern and had included human validation.

**Other Additional Experiments***
We also provided experiments on individual comments raised by reviewers, for example:
**Model Size and Generative Monoculture** In a new experiment in Appendix G, we answer questions around how model size impacts generative monoculture. We recreate our experiments around sentiment in book reviews with Llama 70b (instruction fine-tuned + aligned, but quantized due to the duration of the rebuttal period) and compare to LLama-13-b: we find that while larger models have slightly more diverse output than smaller models, but that they still demonstrate generative monoculture.  Over 50% of generations in LLama 70b were in the most positive sentiment bucket (0.95-1.00 sentiment) for both prompting strategies from the original paper, whereas no one sentiment bucket had such a large majority in the src data.

***Impact of Length and Perplexity Thresholding on Results*** We also investigated if our choice to standardize book reviews by keeping them between 300-700 words, or not considering LLM generations over a certain threshold of perplexity, influenced the severity of generative monoculture in the paper. We found that neither of these have a large impact: the distribution of sentiment is similar over book reviews in the range of 300-700 words and other lengths as we show in Appendix H, and including high-perplexity LLM outputs changes the average sentiment scores on the magnitude of 0.001, meaning that the impact of filtering on the sentiment is negligible. We also demonstrate in Appendix C.1 that the generations over a certain perplexity thresholds are unusable output, which is why we did not include them.

***Data validity experiment*** We curated a new dataset by filtering books published after October 1, 2023, from the GoodReads dataset (May 2024 version); this is to ensure that there is no contamination. Of 79 books found, 72 had exactly 5 reviews, forming a dataset of 360 samples with varied sentiment scores. Using this, we instruction-tuned Llama-13b (pre-trained and chat versions) with parameter-efficient fine-tuning (LoRA dimension=4) over 3 epochs, ensuring minimal perplexity increase on "wikitext-2-raw-v1" to avoid overfitting. Sentiment analysis showed fine-tuning the pre-trained model slightly increased positivity while the chat model skewed more negative, revealing RLHF significantly impacts sentiment despite fine-tuning. This suggests that monoculture is not an artifact of the broader (unseen) distribution (in reference to W.1-2 raised by reviewer L9QX). Results are detailed in Appendix I.

---

> ### Author Response · Authors · 2024-11-22
> **(part 2)**
>
> ***Comments on Dataset Size + Cost*** We also note some reviewers had comments on the size of our coding dataset— we clarify that regarding the set of 100 problems, this means we generated and autojudged 10,000 solutions (100 x 100)  in total by GPT-4 (and 20,000 solutions in total by Claude-3) on these 100 problems (described in “Generation and correctness evaluation of code solutions”). Moreover, the results on 100 problems already revealed pretty significant trends. It is true that we can aim for a larger dataset, but within a limited budget (these experiments already cost us more than $2,000 in invoking the APIs) and out of consideration for efficient and environmental-friendly experimentation, we strongly believe 100 is a reasonable number.
>
> ***Additional Clarifying Edits to the Paper***
> We have gone ahead and updated several wording changes in the paper (e.g. human generated→ human-written) and how exactly we express our Takeaway #3 in response to reviewer comments. We appreciate the reviewers' help to improve our presentation.
>
> We have also clarified common misunderstandings about aspects of our experimental design, such as how and why we only show the diversity of the distribution of correct code outputs from the LLM. However, we note that several comments about clarity and incompleteness were noted by a reviewer who claimed we did not have an appendix when we did.
>
>
> We hope that our clarifications, additional experiments, and edits answer all of the questions the reviewers had about our approach and results—and ask reviewers to clarify if they have not. We welcome follow-ups and further questions for the remainder of the discussion period.
>
>
> [1] Are Language Models Better at Generating Answers or Validating
> Solutions? https://openreview.net/pdf/d3f724811cb971ff60cd45abfc51bd92d2dd7602.pdf

---

### Comment · Area_Chair_5tKD · 2024-11-25
**Reviewer Response**

Dear Reviewers,

The rebuttal discussion period is coming to a close and the paper currently has a mix of positive and negative reviewers. The authors have spent a lot of time responding to each concern -- can you take a look at the author responses and let them know any remaining concerns you have?

Best,
AC

---

### Meta-Review · Area_Chair_5tKD · 2024-12-23

**Metareview:**

Quoting a reviewer’s summary of the paper’s main idea -- “The main idea of this paper can be summarized in the following toy example: if 90% of humans say that chocolate is tasty, should language models always describe chocolate as tasty, or should they aim to somehow reflect the diversity in human opinions, and occasional refer to chocolate as bad-tasting?” The authors study this diversity in two domains: the sentiment of book reviews and the algorithms employed in generated code. The work analyzes the various ways in which LLMs enforce a monoculture by failing to model the diversity present in actual human-written content, and argues that it seems to get worse for models which are “aligned” to human preferences via RLHF. The problem of “generative monoculture” in LLMs is interesting (though not completely new). The results/discussion presented in the paper is thorough and convincing, and the paper is well-written. One of the key concerns is that the evaluation is mostly based on automatic metrics, such as using GPT-3.5 for evaluating LLM outputs. It’s unclear how reliable GPT-3.5 could be as a judge (though the authors presented certain studies of the judgement quality).

**Additional Comments On Reviewer Discussion:**

The discussion seems effective as a few reviewers increased their scores after the discussion. The authors responded to each of reviewers' comments. The main concern left is that the evaluation is mostly based on automatic metrics, such as using GPT-3.5 for evaluating LLM outputs. It’s unclear how reliable GPT-3.5 could be as a judge. There is another concern of regarding attribution of monoculture to training data, which I think is relatively minor. I lean toward acceptance of the work given that it's an interesting quantitative study of “generative monoculture” in LLMs and presents a bunch of related results/discussion.

---

### Decision · Program_Chairs · 2025-01-22

Accept (Poster)